# Hierarchical ODE: Learning Continuous-Time Physical Prototypes for Early Link Failure Detection

**Jiaen Lv** [1]  **Leran Qi** [1]  **Shaowei Wang** [1]

## Abstract

Time series prototype learning is fundamentally challenged by observational ambiguity. Discrete architectures fail to resolve this, as they lack the capacity to decouple stochastic noise from continuous dynamics. Furthermore, rigid closed-set assumptions fail to capture unseen diversity. To address these limitations, we propose a hierarchical ordinary differential equation clustering network, which utilizes neural ordinary differential equation to model latent state evolution as a continuous integral curve. This formulation enforces temporal continuity to effectively disentangle smooth feature trends from stochastic noise, while our adaptive hierarchical mechanism autonomously determines the appropriate number of prototypes without rigid prior constraints. Validated on the early link failure detection task with irregularly sampled time series, the proposed method effectively extracts underlying physical prototypes, thereby enabling robust failure detection. Our code is available at https://github.com/NJ-LNN/Hierarchical-ODE.

## 1. Introduction

Time series prototype learning has emerged as a paradigm for analyzing complex dynamical systems, offering a framework to distill high-dimensional temporal data into interpretable latent representations (Ye & Keogh, 2009). These representations serve as a semantic bridge linking low-level observations to high-level interpretations (Chen et al., 2019). This approach is critical in next-generation wireless communication networks (Letaief et al., 2019; Zhu & Wang, 2023). In such environments, raw signal series are inherently corrupted by stochastic fluctuations (Tao & Wang,

2022). These perturbations introduce feature ambiguity and obscure structural differences between diverse degradation patterns, resulting in the observational entanglement of disparate signal prototypes (Higgins et al., 2017; Tonekaboni et al., 2021). Consequently, the primary challenge lies in mapping these ambiguous, non-stationary signal series to distinct representation patterns that correspond to specific physical prototypes, extracting dynamic features invariant to stochastic noise. Among various network management tasks, the proactive handover decision serves as a representative application scenario (Ak & Canberk, 2021). This task functions as early link failure detection, aiming to predict imminent link interruptions to facilitate timely resource reallocation. Since such interruptions are manifested through signal attenuation, distinguishing the underlying mechanisms governed by continuous dynamics, rather than simply monitoring instantaneous signal intensity, is essential to optimize connection reliability (Eldele et al., 2021).

The primary practical obstacle to reliable early link failure detection lies in the confounding observational patterns exhibited by distinct physical degradation prototypes within the raw signal series (Krueger et al., 2021). This ambiguity is particularly pronounced in rapid signal attenuation, where superficially identical signal series can arise from fundamentally divergent physical prototypes, a phenomenon formally described as spurious correlation (Creager et al., 2021). For instance, an abrupt decline in intensity could signify a persistent degradation prototype (e.g., entering an elevator), necessitating an immediate handover to prevent link failure, or merely a transient fluctuation prototype (e.g., shadowing by a moving obstacle), where the signal integrity is expected to recover. Recent studies in disentangled representation learning have highlighted the necessity of separating such trend-based dynamics from transient variations (Woo et al., 2022; Träuble et al., 2021). Lacking the capacity to disentangle these disparate physical prototypes from ambiguous signal observations, the system risks misinterpreting the network state, leading to either missed detection of actual failures or unnecessary handovers during temporary disturbances (Van Amersfoort et al., 2020).

Nevertheless, existing methods largely rely on discrete architectures, such as recurrent neural network (RNN) and long

---

[1]School of Electronic Science and Engineering, Nanjing University, Nanjing, China. Correspondence to: Shaowei Wang <wangsw@nju.edu.cn>.

*Proceedings of the 43$^{rd}$ International Conference on Machine Learning*, Seoul, South Korea. PMLR 306, 2026. Copyright 2026 by the author(s).

short-term memory (LSTM). While effective for discrete-time prediction, these models impose discretization on continuous physical processes (Rubanova et al., 2019; Kidger et al., 2020). By reducing these processes to discrete state transitions, they function primarily as discrete sequence approximators, mechanisms that interpolate discrete observations based on statistical dependencies, thereby neglecting the derivative function driving the process (Rusch & Mishra, 2021). This inability precludes them from serving as continuous vector field learners, architectures designed to learn the differential equations governing system dynamics (Greydanus et al., 2019; Hasani et al., 2021). Consequently, discrete models fail to capture these governing dynamics, leaving them unable to capture the differential properties required to resolve the aforementioned prototypes.

To alleviate these discretization constraints, contemporary methodologies have prioritized the integration of explicit alignment mechanisms and structural priors. Notable examples include (Hu et al., 2025), which constructs a static metric space to resolve geometric dependencies, and (Chen et al., 2025), which employs frequency-domain attention to align periodic features. Drawing further inspiration from linguistic syntax trees, (Opper & N, 2025) introduces hierarchical constraints to enhance representation efficiency. Despite demonstrating efficacy in static regression or periodic analysis, these strategies, along with recent multi-view imputation frameworks (Liu et al., 2024) and static visual prototype learning (Silva et al., 2025), fail to incorporate continuous evolution modeling. By confining their focus to the alignment of discrete observations or the delineation of static boundaries, such architectures inherently neglect the continuous differential dynamics driving the signal. This theoretical limitation consequently hinders the capability to capture causal mechanisms underlying non-periodic transient events, exemplified by sudden signal interruptions due to occlusion.

In contrast, neural ordinary differential equation (ODE) (Chen et al., 2018) addresses these limitations by modeling the continuous physical processes through a continuous vector field. Rather than approximating transitions between discrete steps, the neural ODE framework formulates the latent state evolution as an integral curve defined by the learnable derivative function. This integral-based formulation is critical, it enforces temporal continuity on the generated trajectory. By adhering to the learned derivatives, the model acts as a natural stabilizer that effectively decouples the smooth evolution of persistent degradation from the irregular noise of transient fluctuations. Consequently, by recovering the differential equation itself, the model accurately disentangles the latent physical prototypes obscured by stochastic perturbations.

Beyond the challenge of modeling continuous physical pro-

cesses, a critical obstacle lies in unsupervised prototype discovery within open-world environments (Vaze et al., 2022; Han et al., 2021). In practical scenarios, signal degradation adheres to specific physical prototypes rather than random occurrences. For instance, entering an elevator triggers abrupt cutoff dynamics, whereas moving towards a cell edge exhibits gradual decay characteristics. Differentiating these prototypes is essential for characterizing signal attenuation behaviors and distinguishing immediate outages from slow fading, processes that dictate the optimal timing for proactive handovers. Nevertheless, the diversity of these prototypes cannot be predetermined. Existing deep clustering frameworks typically rely on a closed-set assumption necessitating a predefined cluster count. This rigidity is inadequate for discovering latent physical prototypes, as an incorrect preset inevitably leads to the merging of disparate signal patterns or the fragmentation of a single prototype (Ronen et al., 2022).

To bridge these gaps, we propose a hierarchical ODE clustering network as a unified framework for continuous-time prototype discovery and early link failure detection. Specifically, we leverage neural ODE to parameterize latent state derivatives and enable the reconstruction of continuous trajectories from sparse signal series leading up to link failure. By formulating the latent evolution as an integral curve, this continuous-time inductive bias acts as a natural stabilizer that decouples smooth degradation trends from stochastic fluctuations, effectively preserving the differential properties required to distinguish complex physical prototypes. Addressing the open-world challenge, we introduce a hierarchical prototype discovery module within the latent space. By analyzing the structure of latent trajectories and employing a dynamic cut-off strategy, the method adaptively determines the appropriate number of prototypes, thereby circumventing the limitations of a predetermined cluster count. For the early link failure detection, the proposed framework matches the emerging trend of the signal against the discovered specific physical prototypes, thereby distinguishing actual degradation from transient fluctuations at an early stage, facilitating proactive handover decisions before a complete link failure occurs. We validate the framework on the proactive network handover task, a scenario governed by continuous physical dynamics yet characterized by noisy and irregular measurements. Our main contributions are summarized as follows.

- We propose a continuous-time representation learning framework that models the continuous latent dynamics via neural ODE. By formulating the evolution as an integral curve, the method effectively reconstructs robust system dynamics from irregular and noisy signal series, surpassing discrete-time baselines in capturing differential properties.

- We introduce an adaptive open-set prototype discovery mechanism. By employing a hierarchical structure with a dynamic cut-off strategy, the model autonomously determines the number of latent physical prototypes, eliminating the dependency on a predetermined cluster count and resolving the ambiguity of unknown physical prototypes.

- We validate the framework on the real-world proactive network handover task. By executing real-time matching between streaming signal inputs and the discovered physical prototypes, the system identifies when emerging attenuation patterns align with a specific degradation prototype. Experimental results demonstrate that our framework effectively differentiates persistent degradation from transient fluctuations, enabling early link failure detection and shifting the decision paradigm from reactive monitoring to proactive anticipation.

## 2. Problem Formulation

In this section, we formalize the framework for analyzing irregular time series. The problem is decomposed into two stages: unsupervised prototype learning, which involves feature extraction and open-set clustering, and early link failure detection, which utilizes the learned prototypes for real-time inference.

### 2.1. Unsupervised Prototype Learning

The objective of this stage is to map raw, irregular sensor streams into interpretable latent clusters without supervision. This process consists of two sub-tasks: continuous-time feature learning and open-set prototype discovery.

**Continuous Feature Learning.** Let $\mathcal{D} = \{\mathcal{S}^{(n)}\}_{n=1}^N$ denote a dataset consisting of $N$ time series samples, where each sequence captures the signal evolution within a specific observation window immediately preceding a link failure event. Unlike standard discrete time series, each sample is represented as a $\mathcal{S} = \{(\mathbf{x}_i, t_i)\}_{i=1}^L$. Here, $\mathbf{t} = [t_1, t_2, \ldots, t_L] \in \mathbb{R}^L$ represents the sequence of continuous timestamps, where the time intervals $\Delta t_i = t_{i+1} - t_i$ are non-uniform and stochastic. $\mathbf{x}_i \in \mathbb{R}^{d_x}$ represents the observation vector at timestamp $t_i$.

The primary goal is to learn a mapping function $f_\theta : \mathcal{S} \to \mathbf{z}$ that projects the variable-length irregular sequence into a fixed-dimensional latent representation. This representation must capture the differential properties of the underlying physical process, remaining robust to sampling irregularity and noise.

**Open-Set Prototype Discovery.** Upon obtaining the latent representations, the objective is to discover a set of canonical patterns, defined as Prototypes $\mathcal{P} = \{\mathbf{p}_k\}_{k=1}^K$, where $\mathbf{p}_k \in \mathbb{R}^{d_z}$ represents the centroid of the $k$-th physical mode in the latent space.

A critical constraint in this work is the open-set setting, where the true number of underlying patterns $K$ cannot be predetermined. Consequently, the task requires simultaneously determining the cluster count $K$ and learning the prototype parameters $\mathcal{P}$ such that they appropriately represent the distinct dynamical modes inherent in the data distribution, avoiding both the over-segmentation of single prototypes and the merging of disparate patterns.

### 2.2. Early Link Failure Detection

Focusing on online detection, we match the emerging time series to learned degradation prototypes. This matching process aims to pinpoint the specific degradation pattern within the ongoing observations, distinguishing imminent failure risks from transient fluctuations to facilitate proactive handover decisions prior to connectivity loss.

## 3. Methodology

**Structural Incapacity of Discrete Architectures.** While standard RNN provides a foundational approach for sequence modeling, they inherently treat time as a sequence of discrete index steps. A standard update $\mathbf{h}_i = \text{RNNCell}(\mathbf{x}_i, \mathbf{h}_{i-1})$ implies that the latent state either remains static or undergoes a simplistic decay during the variable interval $\Delta t = t_i - t_{i-1}$. Consequently, discrete architectures lack the representational capacity to capture the continuous latent dynamics.

**Modeling Continuous Dynamics with Neural ODE.** To overcome this deficiency, we integrate neural ODE (Chen et al., 2018; Rubanova et al., 2019). Formally, Neural ODE parametrizes the continuous dynamics of a hidden state $\mathbf{h}(t)$ using a neural network $f_\theta$. The evolution of the state is defined as an initial value problem (IVP):

$$\frac{d\mathbf{h}(t)}{dt} = f_\theta(\mathbf{h}(t), t), \tag{1}$$

where $f_\theta$ learns the underlying vector field. Given a state $\mathbf{h}(t_0)$ at time $t_0$, the state at any future time $t$ is computed via numerical integration:

$$\mathbf{h}(t) = \mathbf{h}(t_0) + \int_{t_0}^t f_\theta(\mathbf{h}(\tau), \tau)d\tau. \tag{2}$$

This formulation introduces a strong inductive bias towards modeling the intrinsic physical dynamics, allowing the network to infer unobserved latent evolution and naturally bridge the irregular time gaps.

### 3.1. Continuous-Time Autoencoder Framework

Our proposed framework consists of a continuous-time encoder that alternates between dynamic evolution and observation injection, followed by a generative ODE decoder.

#### 3.1.1. CONTINUOUS-TIME ENCODING VIA NEURAL ODE

Let the irregular time series be denoted as $\mathcal{S} = \{(\mathbf{x}_i, t_i)\}_{i=1}^{L}$, where $\mathbf{x}_i \in \mathbb{R}^{d_x}$ represents the observation vector at timestamp $t_i$. Unlike standard recurrent models that operate on fixed indices, our encoder processes the signal on a continuous time axis, synergizing autonomous evolution with measurement updates.

**Dynamics Evolution.** Between two consecutive timestamps $t_{i-1}$ and $t_i$, the system undergoes physical evolution governed by the learned vector field. We model this process using a neural ODE. Let $\Delta t_i = t_i - t_{i-1}$ be the time interval. The latent vector $\mathbf{h}(t) \in \mathbb{R}^{d_h}$ evolves from the previous timestamp by solving the IVP:

$$\mathbf{h}(t_i^-) = \mathbf{h}(t_{i-1}) + \int_{t_{i-1}}^{t_i} f_\theta(\mathbf{h}(\tau), \tau)d\tau, \qquad (3)$$

where $f_\theta$ is a neural network parameterizing the derivative function. This formulation captures the intrinsic instantaneous rate of change of the physical signal.

**Observation-Driven Rectification.** At timestamp $t_i$, the continuous evolution is interrupted to incorporate the new external evidence $\mathbf{x}_i$. We employ a gated recurrent unit (GRU) mechanism to adjust the latent state, formulated as follows,

$$\mathbf{r}_i = \sigma(\mathbf{W}_r[\mathbf{h}(t_i^-), \mathbf{x}_i] + \mathbf{b}_r),$$
$$\mathbf{z}_i = \sigma(\mathbf{W}_z[\mathbf{h}(t_i^-), \mathbf{x}_i] + \mathbf{b}_z),$$
$$\mathbf{n}_i = \tanh(\mathbf{W}_n[\mathbf{r}_i \odot \mathbf{h}(t_i^-), \mathbf{x}_i] + \mathbf{b}_n),$$
$$\mathbf{h}(t_i) = (1 - \mathbf{z}_i) \odot \mathbf{n}_i + \mathbf{z}_i \odot \mathbf{h}(t_i^-), \qquad (4)$$

where $\mathbf{W}$ and $\mathbf{b}$ denote the learnable weights and biases. The terms $\mathbf{r}_i$, $\mathbf{z}_i$, and $\mathbf{n}_i$ represent the reset gate, update gate, and candidate activation, respectively, which control the information fusion.

Crucially, this update step defines the relationship between the predicted and corrected states. $\mathbf{h}(t_i^-)$ serves as the evolved prior, representing the system's state derived purely from the underlying physical laws integrated up to $t_i$. Upon receiving $\mathbf{x}_i$, the recurrent unit induces a discrete jump to produce the rectified posterior $\mathbf{h}(t_i)$. This iterative process accumulates the historical information into the latent state. Upon processing the entire sequence, the final hidden state $\mathbf{h}(t_L)$ is extracted as the fixed-length latent embedding, denoted as $\mathbf{z}_{lat}$. Unlike the encoder which relies on observational rectifications, the decoding process is purely generative, governed solely by the learned physical dynamics without external intervention.

#### 3.1.2. GENERATIVE DECODING AND TRAJECTORY RECONSTRUCTION

Upon processing the entire sequence, the final hidden state serves as the fixed-length latent embedding, denoted as $\mathbf{z}_{lat}$. Unlike the encoder which relies on observational rectifications, the decoding process is purely generative, governed solely by the learned physical dynamics without external intervention.

**Trajectory Reconstruction.** The learned embedding $\mathbf{z}_{lat}$, derived from the final state of the encoder, serves as the terminal boundary condition $\mathbf{h}(t_L)$ for the decoding process. To reconstruct the historical signal dynamics over the observed time window $\mathcal{T} = [t_0, t_L]$, we solve the IVP backward in time using the shared neural vector field $f_\theta$. Formally, given $\mathbf{h}(t_L) = \mathbf{z}_{lat}$, the trajectory is recovered via reverse-time integration:

$$\hat{\mathbf{h}}(t) = \text{ODESolve}(f_\theta, \mathbf{z}_{lat}, t), \quad \forall t \in \{t_i \in \mathcal{T} \mid t_i \leq t_L\}, \qquad (5)$$

where the solver integrates from $t_L$ back to $t_0$. Crucially, this continuous backward generation is independent of the irregular sampling grid, producing a smooth trajectory $\hat{\mathbf{h}}(t)$ that inherently interpolates between observations. This mechanism ensures that the reconstructed dynamics strictly adhere to the learned physical conservation laws, effectively recovering the continuous evolution history from the terminal latent representation.

**Signal Reconstruction.** The generated latent states $\hat{\mathbf{h}}(t)$ represent the system's internal physical evolution on the continuous manifold. To map these latent dynamics back to the observable data space, we employ a learnable decoding function $g_\phi$ (e.g., a Multilayer Perceptron):

$$\hat{\mathbf{x}}(t) = g_\phi(\hat{\mathbf{h}}(t)). \qquad (6)$$

The reconstruction objective aims to minimize the discrepancy between the recovered trajectory and the ground-truth observations. We define the loss function as the mean squared error (MSE) over all available timestamps:

$$\mathcal{L} = \frac{1}{L} \sum_{i=1}^{L} \|\mathbf{x}_i - \hat{\mathbf{x}}(t_i)\|_2^2. \qquad (7)$$

By minimizing $\mathcal{L}$, the framework forces the learned ODE dynamics to capture the dominant physical trends of the signal. Unlike discrete models that may overfit to high-frequency jitter, this generative process prioritizes the recovery of the smooth underlying function, ensuring that the reconstructed signal reflects the true physical behavior governing the observations.

## 3.2. Two-Stage Adaptive Prototype Discovery

The continuous-time encoder maps each irregular time series to the fixed-dimensional latent space $\mathcal{Z} \subset \mathbb{R}^{d_h}$. Specifically, by aggregating the latent embeddings $\mathbf{z}_{lat}$ derived from all $N$ sequences in the dataset, we construct the global latent representation matrix $\mathbf{Z} \in \mathbb{R}^{N \times d_h}$.

To extract robust and interpretable representations of the underlying system modes from $\mathbf{Z}$, we propose a two-stage clustering strategy. This approach systematically combines the structural discovery capability of agglomerative hierarchical clustering with the optimization efficiency of K-Means to generate high-fidelity prototypes.

**Hierarchical Initialization.** The first stage focuses on identifying the appropriate number of prototypes without rigid prior assumptions. We perform agglomerative hierarchical clustering on the latent matrix $\mathbf{Z}$ to uncover the multi-scale manifold structure. To ensure the generation of compact, spherical clusters suitable for prototype representation, we utilize Ward minimum variance method as the linkage criterion. In each iteration, the algorithm merges two clusters $C_a$ and $C_b$ that result in the minimal increase in total within-cluster variance. The distance metric for merging is formally defined as:

$$d(C_a, C_b) = \frac{|C_a| \cdot |C_b|}{|C_a| + |C_b|} \|\boldsymbol{\mu}_a - \boldsymbol{\mu}_b\|_2^2, \qquad (8)$$

where $\boldsymbol{\mu}_a$ and $\boldsymbol{\mu}_b$ denote the centroids of the respective clusters, and $|\cdot|$ represents the cluster size. This process constructs a dendrogram that encapsulates the hierarchical relationships among data points.

Determining the appropriate cut in the dendrogram to derive the cluster count $K$ is a critical step. A fixed global threshold often fails to adapt to the varying densities of the high-dimensional feature space. To address this, we introduce a locally-constrained threshold search strategy. Let $\tau_{base}$ denote a hyperparameter representing the expected scale of semantic similarity. We define a local search interval $\Omega = [\tau_{base} - \epsilon, \tau_{base} + \epsilon]$ and seek a robust threshold $\tau^*$ within this range. The objective is to find a cut that yields the most concise representation (i.e., the minimal number of clusters) while strictly adhering to the similarity constraints imposed by the interval. The number of clusters $K^*$ is determined by:

$$K^* = \min_{\tau \in \Omega} \left( \text{Count}(\text{TreeCut}(\mathbf{Z}, \tau)) \right). \qquad (9)$$

This strategy effectively filters out redundant micro-clusters and identifies the dominant structural components of the data, providing a structurally sound $K^*$ for the subsequent refinement.

While hierarchical clustering effectively identifies the primary structural count $K^*$, utilizing a single centroid to represent a complex, non-convex cluster often results in suboptimal positioning. Therefore, in the second stage, we employ localized K-Means clustering to discover fine-grained prototypes within each of the $K^*$ coarse clusters identified by the first stage.

For each coarse cluster $\mathcal{G}_k$ (where $k = 1, \ldots, K^*$), we apply K-Means to identify a set of sub-prototypes $\{\boldsymbol{\mu}_{k,j}\}_{j=1}^{P_k}$ that collectively represent the cluster's internal distribution. The algorithm minimizes the intra-cluster inertia by optimizing these sub-prototypes:

$$\mathcal{J} = \sum_{k=1}^{K^*} \sum_{j=1}^{P_k} \sum_{\mathbf{z}_i \in \mathcal{S}_{k,j}} \|\mathbf{z}_i - \boldsymbol{\mu}_{k,j}\|_2^2, \qquad (10)$$

where $P_k$ denotes the number of sub-prototypes within the $k$-th coarse cluster, and $\mathcal{S}_{k,j}$ represents the subset of samples assigned to the $j$-th sub-prototype $\boldsymbol{\mu}_{k,j}$. Upon convergence, the aggregate collection of all sub-prototypes $\{\boldsymbol{\mu}_{k,j}\}$ constitutes the final dictionary of the system's dynamic prototypes, allowing each physical class $k$ to be represented by multiple distinct prototypes to capture its complex geometry.

**Prototype Characterization.** Finally, to fully characterize the spatial extent of each prototype for tasks, we define a coverage radius $R_{k,j}$ for each sub-prototype as the distance from the sub-centroid to the farthest sample assigned to that specific sub-component,

$$R_{k,j} = \max_{\mathbf{z} \in \mathcal{S}_{k,j}} \|\mathbf{z} - \boldsymbol{\mu}_{k,j}\|_2. \qquad (11)$$

This rigorous definition ensures that each sub-prototype strictly encompasses the observed local variations within its specific region. The resulting hierarchical memory structure $\mathcal{P} = \{\{(\boldsymbol{\mu}_{k,j}, R_{k,j})\}_{j=1}^{P_k}\}_{k=1}^{K^*}$ constitutes the learned knowledge base, where the $k$-th physical pattern is comprehensively defined by the union of its $P_k$ constituent sub-regions. The complete training pipeline and the subsequent adaptive prototype discovery process are summarized in Algorithm 1.

## 3.3. Online Classification and Inference

In the deployment phase, the framework continuously monitors the incoming irregular signal stream via an online sliding window of length $T$. For a given observation sequence $\mathcal{S}$, the continuous-time encoder computes the terminal latent state $z_T$. The hierarchical clustering module provides a library of physical prototypes, each explicitly parameterized by a centroid $\mu_{kj}$ and a maximum tolerance radius $R_{kj}$, which serve as the geometric decision boundaries. We mathematically define the decision function $D(\mathcal{S})$ as:

$$D(\mathcal{S}) = \begin{cases} 1 \text{ (degrading)}, & \text{if } \exists(k, j) \text{ such that} \\ & \|z_T - \mu_{kj}\| \leq R_{kj}, \\ 0 \text{ (stable)}, & \text{otherwise.} \end{cases} \qquad (12)$$

A decision of $D(\mathcal{S}) = 1$ indicates that the latent trajectory has fallen into a known prototype region, thereby triggering a proactive handover. In complex multipath environments where the tolerance boundaries of distinct prototypes occasionally intersect, the system resolves overlaps by assigning the latent state to the nearest prototype based on the minimal Euclidean distance: $\arg\min_{k,j} \|z_T - \mu_{kj}\|$.

To strictly quantify the robustness of this decision mechanism, we evaluate the system on a stable set $S_{stable}$ with steady connections and a degrading set $S_{degrade}$ with signal decay preceding link failure. The performance is governed by two critical metrics: the False Acceptance Rate (FAR) and the False Rejection Rate (FRR), formulated as:

$$
\begin{aligned}
\text{FAR} &= \frac{\sum_{\mathcal{S} \in S_{stable}} D(\mathcal{S})}{|S_{stable}|}, \\
\text{FRR} &= \frac{\sum_{\mathcal{S} \in S_{degrade}} (1 - D(\mathcal{S}))}{|S_{degrade}|}.
\end{aligned}
\tag{13}
$$

Minimizing the FAR prevents unnecessary handovers caused by transient signal jitter, while minimizing the FRR ensures that impending link failures are proactively anticipated.

## 4. Numerical Results

In this section, we empirically validate the proposed framework on the real-world task of proactive network handover. To comprehensively assess the efficacy of the proposed framework, we conduct a systematic evaluation across three critical dimensions: the **performance of early link failure detection**, the **visual integrity of the learned clusters**, and the model's **reconstruction error and robustness to sparsity**.

### 4.1. Experimental Setup

We utilize a real-world dataset collected from a high-density CBD office building, characterized by hybrid WiFi/cellular infrastructures and significant structural obstructions. Data spans six scenarios (offices, corridors, staircases, meeting rooms, restrooms, tea rooms) with unique multi-path characteristics inducing fundamentally different degradation prototypes. Complementing these weak-coverage scenarios, we also collect trajectories from areas with stable network coverage. To reflect real-world constraints, we constructed a small-sample dataset (20–30 sequences per scenario) of received signal strength indicator (RSSI) trajectories recorded along multiple converging routes, thereby capturing diverse signal degradation prototypes.

To rigorously isolate the impact of continuous-time modeling, we compare the proposed ODE method against discrete baseline variants. In these baselines, the core ODE component within the proposed framework is replaced by stan-

---

**Algorithm 1** Continuous-Time Prototype Learning Framework

1: **Input:** Dataset $\mathcal{D} = \{\mathcal{S}^{(n)}\}_{n=1}^N$, $\mathcal{S} = \{(\mathbf{x}_i, t_i)\}_{i=0}^L$, and Hyperparameters $\tau_{base}, \epsilon$
2: **Output:** Learned prototypes $\mathcal{P} = \{\{(\boldsymbol{\mu}_{k,j}, R_{k,j})\}_{j=1}^{P_k}\}_{k=1}^{K^*}$, Encoder parameters $\theta$, Decoder parameters $\phi$
3: *// Phase 1: Train ODE-GRU Autoencoder*
4: **while** not converged **do**
5:     Sample mini-batch of sequences from $\mathcal{D}$
6:     Initialize latent state $\mathbf{h}(t_0) = \mathbf{0}$
7:     **for** $i = 1$ **to** $L$ **do**
8:         $\mathbf{h}(t_i^-) = \mathbf{h}(t_{i-1}) + \int_{t_{i-1}}^{t_i} f_\theta(\mathbf{h}(\tau), \tau) d\tau$
9:         $\mathbf{h}(t_i) = \text{GRUCell}(\mathbf{x}_i, \mathbf{h}(t_i^-))$
10:     **end for**
11:     Get latent embedding $\mathbf{z}_{lat} = \mathbf{h}(t_L)$
12:     *Generative Decoding:*
13:     Solve ODE: $\hat{\mathbf{h}}(t) = \text{ODESolve}(f_\theta, \mathbf{z}_{lat}, t)$
14:     Reconstruct: $\hat{\mathbf{x}}(t) = g_\phi(\hat{\mathbf{h}}(t))$
15:     Update $\theta, \phi$ by minimizing loss: $\mathcal{L} = \frac{1}{L} \sum_{i=1}^L \|\mathbf{x}_i - \hat{\mathbf{x}}(t_i)\|_2^2$
16: **end while**
17: *// Phase 2: Adaptive Prototype Discovery*
18: Extract embeddings $\mathbf{Z} \in \mathbb{R}^{N \times d_h}$ for all samples using trained Encoder
19: **Step 1: Structure Identification**
20: Perform Hierarchical Clustering on $\mathbf{Z}$ with Ward's linkage
21: Search threshold $\tau^* \in [\tau_{base} - \epsilon, \tau_{base} + \epsilon]$
22: Determine cluster count $K^* = \text{Count}(\text{TreeCut}(\mathbf{Z}, \tau^*))$
23: **Step 2: Prototype Refinement**
24: Initialize centroids using hierarchical results
25: **for** $k = 1$ **to** $K^*$ **do**
26:     Refine cluster $\mathcal{C}_k$ into sub-prototypes $\{\boldsymbol{\mu}_{k,j}\}_{j=1}^{P_k}$ via localized K-Means
27:     Compute coverage radii: $R_{k,j} = \max_{\mathbf{z} \in \mathcal{S}_{k,j}} \|\mathbf{z} - \boldsymbol{\mu}_{k,j}\|_2$
28: **end for**
29: **Return** Prototypes $\mathcal{P} = \{\{(\boldsymbol{\mu}_{k,j}, R_{k,j})\}_{j=1}^{P_k}\}_{k=1}^{K^*}$

---

dard discrete temporal encoders, specifically LSTM, GRU, ResNet and ModernTCN (Luo & Wang, 2024). This experimental design ensures a fair comparison by maintaining a consistent architectural topology while varying only the mechanism for latent state evolution.

### 4.2. Performance of Early Link Failure Detection

We achieve early detection by matching emerging precursor time series to learned physical prototypes within the strictly defined pre-failure window. Since all discovered clusters correspond to degradation prototypes, matching

*Table 1.* Performance Comparison across Six Scenarios.

| Method | Scenario 1 | | Scenario 2 | | Scenario 3 | | Scenario 4 | | Scenario 5 | | Scenario 6 | |
|---|---|---|---|---|---|---|---|---|---|---|---|---|
| | FAR | FRR | FAR | FRR | FAR | FRR | FAR | FRR | FAR | FRR | FAR | FRR |
| ResNet | $0.04 \pm 0.01$ | $0.82 \pm 0.07$ | $0.04 \pm 0.01$ | $0.66 \pm 0.15$ | $0.03 \pm 0.01$ | $0.93 \pm 0.06$ | $0.04 \pm 0.01$ | $0.75 \pm 0.14$ | $0.03 \pm 0.01$ | $0.93 \pm 0.07$ | $0.04 \pm 0.02$ | $0.96 \pm 0.03$ |
| LSTM | $0.06 \pm 0.03$ | $0.06 \pm 0.06$ | $0.36 \pm 0.09$ | $0.13 \pm 0.12$ | $0.14 \pm 0.07$ | $0.33 \pm 0.09$ | $0.05 \pm 0.01$ | $0.07 \pm 0.09$ | $0.03 \pm 0.01$ | $0.09 \pm 0.10$ | $0.05 \pm 0.04$ | $0.17 \pm 0.16$ |
| GRU | $0.04 \pm 0.02$ | $0.11 \pm 0.10$ | $0.03 \pm 0.02$ | $0.09 \pm 0.10$ | $0.01 \pm 0.02$ | $0.29 \pm 0.19$ | $0.04 \pm 0.04$ | $0.12 \pm 0.11$ | $0.04 \pm 0.01$ | $0.14 \pm 0.07$ | $0.03 \pm 0.01$ | $0.32 \pm 0.21$ |
| ModernTCN | $0.04 \pm 0.01$ | $0.33 \pm 0.17$ | $0.04 \pm 0.01$ | $0.54 \pm 0.21$ | $0.04 \pm 0.01$ | $0.61 \pm 0.12$ | $0.04 \pm 0.02$ | $0.22 \pm 0.14$ | $0.03 \pm 0.01$ | $0.33 \pm 0.14$ | $0.04 \pm 0.01$ | $0.94 \pm 0.05$ |
| **ODE (Ours)** | $0.04 \pm 0.01$ | $0.04 \pm 0.06$ | $0.04 \pm 0.02$ | $0.00 \pm 0.01$ | $0.03 \pm 0.01$ | $0.00 \pm 0.01$ | $0.04 \pm 0.03$ | $0.01 \pm 0.00$ | $0.02 \pm 0.03$ | $0.09 \pm 0.07$ | $0.04 \pm 0.01$ | $0.05 \pm 0.08$ |

an incoming sequence to any prototype is classified as a positive detection (failure risk), while trajectories collected from stable coverage zones serve as negative samples. To objectively assess the model capability in balancing security and usability, we utilize two critical metrics, False Acceptance Rate (FAR) and False Rejection Rate (FRR) defined in Section 3.3.

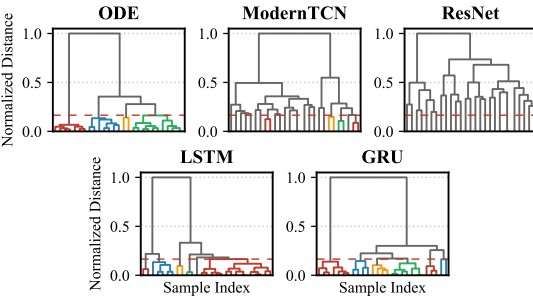

*Figure 1.* Hierarchical Structure of Prototypes. The dendrogram reveals the semantic grouping of latent dynamics, where the red dashed line denotes the adaptive cut-off threshold. Sub-trees merging below this threshold are identified as coherent physical prototypes.

Table 1 reveals a distinct performance pattern across the evaluated degradation scenarios. As observed, all methods, including discrete baselines (LSTM, GRU, ResNet and ModernTCN), achieve relatively low FAR scores. This indicates that most models effectively maintain basic security boundaries, successfully rejecting obvious outliers that deviate significantly from the learned prototypes. This success is attributed to our robust unsupervised framework, where the hierarchical clustering and dynamic thresholding strategy provide a solid baseline for distinguishing disparate clusters.

However, a critical performance divergence emerges in the FRR metric. While discrete baselines maintain security through a low FAR, they exhibit a significantly higher FRR, particularly in scenarios characterized by high noise. This limitation is attributed to their inherent structural constraints, which rigidly approximate dynamics as transitions between fixed discrete steps. Consequently, these models lack the sensitivity to capture continuous evolutionary trends, often misinterpreting the target signal decay as stochastic noise, resulting in the erroneous rejection of valid handover re-

quests.

In contrast, the proposed method achieves the lowest FRR across all scenarios while maintaining a competitive FAR. This superiority arises from the modeling of continuous latent dynamics. By characterizing the signal series through a parameterized vector field, the method extracts robust continuous features that inherently reject stochastic noise, thereby effectively distinguishing the underlying physical evolution from transient noise. Notably, a performance hierarchy is observed among baselines: recurrent models (LSTM, GRU) exhibit lower FRR compared to the ResNet and ModernTCN, yet still lag behind the proposed method. This progression provides insight into the role of temporal modeling. While ResNet and ModernTCN treat time steps as independent feature dimensions, LSTM and GRU incorporate discrete temporal dependencies, offering a coarse approximation of the signal evolution. The progressive improvement from static mapping (ResNet and ModernTCN) to discrete recurrence (RNN) and finally to continuous integration (ODE) validates that continuous-time modeling serves as the decisive factor for capturing the intrinsic dynamics of irregular signal series

### 4.3. Visual Analysis of Cluster Integrity

To further verify the physical consistency of the learned representations, we visualize the hierarchical structure of scenario 1 in Figure 1. To isolate continuous-time modeling gains from clustering bias, we derived $\tau^*$ from the ODE latent space, setting $\tau$ as a fixed percentage of the normalized maximum linkage distance across baselines. The vertical axis quantifies the normalized metric distance between feature vectors, representing the dissimilarity between signal series. The red dashed line indicates the adaptive threshold determined by our dynamic cut-off strategy. This threshold functions as a structural decision boundary: all branches merging below this line are consolidated into a single physical prototype, while connections exceeding this distance are recognized as distinct categories.

A critical divergence is observed in how different models handle signal variations. For discrete baselines (e.g., GRU) in Figure 2, we observe a phenomenon of prototype fragmentation: sequences originating from the same physical event are erroneously split into multiple disjoint sub-clusters. This occurs because discrete models are hypersensitive to

transient fluctuations, where high-amplitude noise shifts the latent representation significantly, causing the model to misinterpret purely stochastic variations as distinct physical patterns.

In contrast, the proposed method successfully maintains cluster integrity, unifying these diverse signal series into a single, coherent cluster in Figure 3. By constraining the latent evolution to follow a smooth integral curve, our method effectively filters out the stochastic perturbations that mislead discrete baselines. This confirms that the continuous-time model can ensure that signals sharing the same governing dynamics are correctly consolidated, regardless of noise intensity.

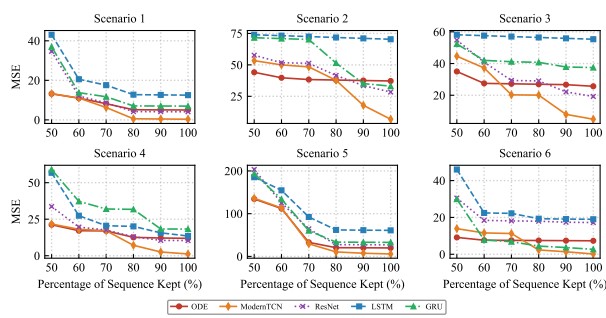

*Figure 4.* Mean Squared Error against Percentage of Sequence Kept. The curves depict the reconstruction error across varying data sparsity levels.

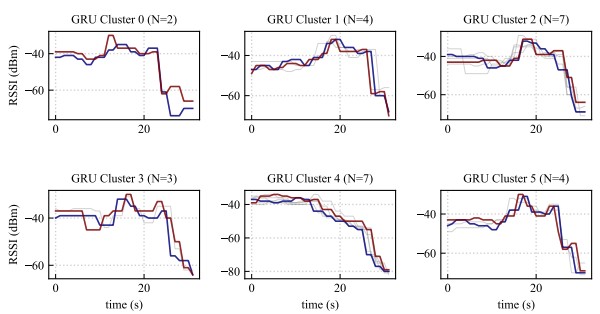

*Figure 2.* Visualization of Signal Series Prototypes for GRU.The figure displays all the signal series assigned to each cluster. The highlighted colored curves represent the corresponding prototypes, capturing the central dynamic trend of each category.

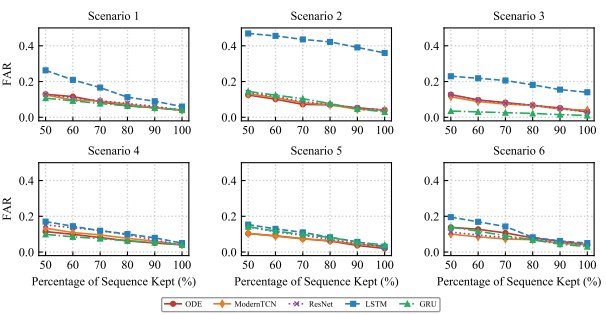

*Figure 5.* FAR against Percentage of Sequence Kept. This evaluation demonstrates the susceptibility to false acceptances across varying levels of signal completeness.

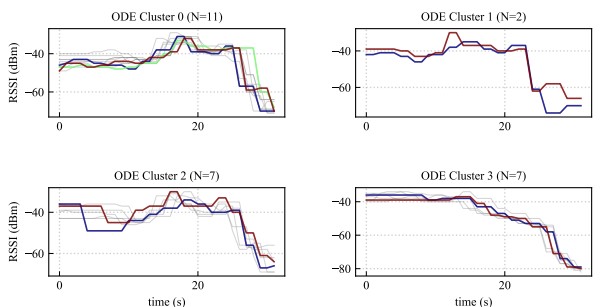

*Figure 3.* Visualization of Signal Series Prototypes for ODE.The figure displays all the signal series assigned to each cluster. The highlighted colored curves represent the corresponding prototypes, capturing the central dynamic trend of each category.

### 4.4. Reconstruction Error and Robustness to Sparsity

To emulate real-world instability where early signal segments may be lost due to connection latency, we employ a delayed-observation evaluation protocol. While models are trained on full sequences, validation is performed on

partial sequences created by discarding the initial segment of the signal series. This effectively evaluates the model ability to reconstruct global dynamics despite the absence of antecedent features. For discrete baselines, we apply nearest-neighbor interpolation to align the remaining timestamps.

As shown in Figure 4, on complete sequences, the proposed method achieves lower MSE than recurrent baselines but slightly higher error than ResNet and ModernTCN. We attribute this to discrete approximators rigidly overfitting high-frequency stochastic noise to minimize point-wise differences. In contrast, our integral curve formulation enforces temporal continuity, naturally filtering out non-informative transient fluctuations to capture the true underlying system dynamics.

This continuous prior confers superior robustness against sparsity. As sequence completeness decreases, the MSE of discrete baselines deteriorates rapidly since they lack the derivative function required to bridge missing intervals. Conversely, our method maintains consistent reconstruction quality. More importantly, this stability is empirically corroborated by the downstream task metrics (Figure 5 and

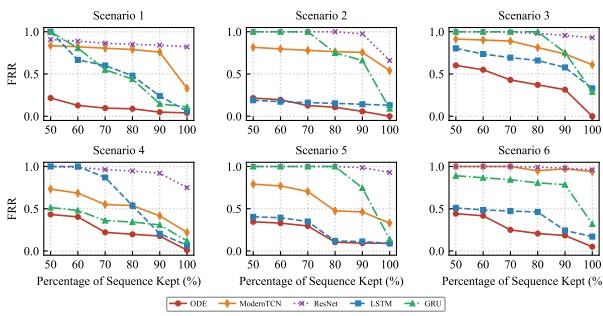

*Figure 6.* FRR against Percentage of Sequence Kept. This evaluation demonstrates the susceptibility to false rejections across varying levels of signal completeness.

Figure 6). While the FAR and FRR of discrete architectures like ModernTCN surge under missing data, our approach maintains stable curves even under extreme sparsity. By effectively extrapolating system dynamics across unobserved time steps, the proposed method proactively anticipates impending link failures (maintaining a low FRR) and filters out transient noise (maintaining a low FAR), guaranteeing robust handover decisions.

### 4.5. Limitations

While the proposed framework demonstrates robust performance in proactive handover tasks, it exhibits specific limitations regarding algorithmic parameterization, out-of-scope data, and prediction horizons. A primary constraint lies in the selection of the optimal distance threshold $\tau^*$. Although the physical scale $\tau$ is more environment-adaptive than predefining a rigid cluster count $K$, determining the exact value of $\tau^*$ still relies on domain heuristics, such as observing the longest stable branch in the hierarchical dendrogram.

A fundamental mathematical assumption of our approach is that the time-series evolution is driven by underlying continuous physical dynamics. Consequently, sequences lacking continuous inertia or dominated by pure discrete logic (e.g., text symbols, high-frequency financial ticks) fall outside our design scope. Applying this continuous-time formulation to such domains is structurally misaligned, as they do not exhibit the smooth differential continuity necessary for the neural ODE prior.

Furthermore, the framework is constrained by its prediction horizon, as the ODE generative decoder encounters limitations during extremely long unobserved windows. Without external observations to trigger the GRU-based state rectification, pure numerical integration accumulates approximation errors, eventually causing trajectory drift. Practical deployment therefore requires intermittent updates within a defined confidence window to periodically recalibrate the latent state and ensure reliable proactive decisions.

## 5. Conclusion

We propose the hierarchical ODE clustering network, a continuous-time framework designed to resolve observational ambiguity in irregular dynamical systems. By parameterizing latent evolution via neural ODE, our method enforces temporal continuity to effectively decouple smooth trends from stochastic noise. Crucially, we integrate adaptive hierarchical clustering with a dynamic cut-off strategy, enabling the autonomous discovery of physical prototypes without reliance on predetermined cluster counts. Experiments on proactive network handover demonstrate that our approach successfully differentiates persistent degradation from transient fluctuations, facilitating reliable early link failure detection.

## Acknowledgments

The authors would like to thank the anonymous reviewers, whose invaluable comments helped improve the presentation of this paper substantially.

## Impact Statement

This paper presents work whose goal is to advance the field of Machine Learning. There are many potential societal consequences of our work, none of which we feel must be specifically highlighted here.

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

# A. Implementation Details

To ensure reproducibility, we provide the detailed architectural specifications for the proposed hierarchical ODE framework and the baseline models.

## A.1. Model Architectures

**Proposed Hierarchical ODE Framework.**   The proposed framework consists of a continuous-time encoder and a generative ODE decoder.

- **Neural Vector Field ($f_\theta$):** The derivative function $f_\theta$, utilized in both the encoder and decoder, is parameterized as a Multi-Layer Perceptron (MLP). It consists of a linear input layer mapping $\mathbb{R}^{d_h+1} \to \mathbb{R}^{2d_h}$ (concatenating the latent state $h(t)$ with the scalar time $t$), followed by a `SiLU` activation function, and a linear output layer mapping $\mathbb{R}^{2d_h} \to \mathbb{R}^{d_h}$.

- **Encoder (Hybrid ODE-GRU):** The encoder employs a hybrid architecture. The continuous evolution between observations is modeled by $f_\theta$ and solved using the fixed-step Runge-Kutta 4 (RK4) method. This choice balances computational efficiency with the short-term integration requirements between frequent observation injections (handled by a standard `GRUCell`).

- **Decoder (Generative ODE):** The decoder is a purely generative ODE rooted at the latent embedding. It utilizes the same network structure for $f_\theta$ but employs the Dormand-Prince (Dopri5) solver with adaptive step sizes. This strategy mitigates error accumulation during the long-horizon trajectory reconstruction, ensuring high fidelity.

For the experimental evaluation, 30% of the collected dataset was reserved as a test set to assess the model generalization performance.

**Baseline Models.**   For a fair comparison, all baselines were configured with comparable latent capacities to the proposed method.

- **Data Preprocessing for Discrete Baselines:** Since standard discrete models (ResNet, LSTM, GRU) assume fixed sampling intervals, we explicitly applied nearest-neighbor interpolation to align the irregular observations to a fixed temporal grid before feeding them into these baselines.

- **ResNet Autoencoder:** Implemented as a 1D ResNet. The encoder comprises an initial convolution ($k = 3, p = 1$) followed by three residual stages. Here, $k$ and $p$ denote the convolution kernel size defining the temporal receptive field and the zero-padding required to preserve sequence length, respectively. Stage 1 contains two residual blocks (16 channels); Stages 2 and 3 contain two residual blocks each (32 and $d_h$ channels, respectively) with a stride of 2 for downsampling. The decoder utilizes a symmetric structure with upsampling layers.

- **Recurrent Baselines (LSTM/GRU):** Both models utilize a bi-directional structure with 2 stacked layers (`num_layers=2`) for both the encoder and decoder to capture complex temporal dependencies.

- **ModernTCN Autoencoder:** Built upon 1D ModernTCN architectures. The encoder maps the input to 32 channels ($d_{model} = 32$) via an initial pointwise convolution ($k = 1$), followed by two ModernTCN blocks. Each block comprises a large-kernel depthwise convolution ($k = 15, p = 7$) to capture temporal dependencies, and a pointwise ConvFFN (expanding to 64 channels with GELU) for channel mixing. Features are then flattened and linearly projected to the latent dimension $d_h = 32$. The decoder mirrors this structure: a linear expansion recovers the sequential feature map, followed by two identical ModernTCN blocks and a final pointwise convolution ($k = 1$) for 1D signal reconstruction.

## A.2. Determination of Clustering Hyperparameters

The hyperparameters for the proposed two-stage adaptive clustering mechanism were empirically determined through a preliminary sensitivity analysis performed on a held-out validation subset (comprising approximately 10% of the training data).

*Table 2.* Computational Complexity Comparison of Core Feature Extraction Architectures.

| Core Architecture | Parameter Complexity | Memory Complexity | Training Time Complexity | Inference Time Complexity |
|---|---|---|---|---|
| LSTM / GRU | $\mathcal{O}(d_h^2 + d \cdot d_h)$ | $\mathcal{O}(L \cdot d_h)$ | $\mathcal{O}(L \cdot d_h^2)$ | $\mathcal{O}(L \cdot d_h^2)$ |
| ModernTCN | $\mathcal{O}(k \cdot d_h + d_h^2)$ | $\mathcal{O}(L \cdot d_h)$ | $\mathcal{O}(L \cdot d_h^2)$ | $\mathcal{O}(L \cdot d_h^2)$ |
| ResNet | $\mathcal{O}(k \cdot d_h^2)$ | $\mathcal{O}(L \cdot d_h)$ | $\mathcal{O}(L \cdot d_h^2)$ | $\mathcal{O}(L \cdot d_h^2)$ |
| **ODE** | $\mathcal{O}(d_h^2 + d \cdot d_h)$ | $\mathcal{O}(d_h)$ | $\mathcal{O}(N \cdot d_h^2)$ | $\mathcal{O}(\tilde{N} \cdot d_h^2)$ |

**Stage 1: Adaptive Threshold Search.** To determine the appropriate cut-off for the hierarchical tree, we employ a bounded search strategy around a base estimate. We define a local search interval $\Omega = [\tau_{base} - \epsilon, \tau_{base} + \epsilon]$ and seek a robust threshold $\tau^*$ within this range. The appropriate $\tau^*$ was selected based on the empirical detection performance on the validation subset, ensuring that the resulting clusters effectively capture distinct degradation patterns without over-fragmentation.

**Stage 2: Sub-Prototype Count for Localized K-Means.** While hierarchical clustering effectively identifies the primary structural count $K^*$, utilizing a single centroid to represent a cluster with high internal diversity often results in suboptimal positioning. Therefore, in the second stage, we employ localized K-Means clustering to discover fine-grained prototypes within each of the $K^*$ coarse clusters identified by the first stage. The number of sub-prototypes for this localized step was calibrated using the validation subset. We evaluated different configurations to identify the setting that best captured the fine-grained characteristics of the coarse clusters. The final parameter was selected to optimize the alignment between the learned prototypes and the validation trajectories, ensuring robust representation of complex signal variations while avoiding unnecessary complexity.

# B. Computational Complexity

To evaluate the computational tradeoffs of the proposed framework, Table 2 summarizes the theoretical complexity across baseline architectures. Let $L$ denote the sequence length, $d_h$ the hidden dimension, $d$ the input dimension, and $k$ the convolution kernel size. For the ODE-based generative model, $N$ and $\tilde{N}$ represent the number of ODE solver evaluations required during the offline training and online inference phases, respectively.

**Parameter Complexity:** All evaluated architectures, including recurrent networks (LSTM/GRU) and convolutional baselines (ResNet, ModernTCN), alongside the proposed ODE framework, share a parameter complexity bounded by $\mathcal{O}(d_h^2)$. This parity ensures that the observed performance gains stem fundamentally from the continuous-time modeling paradigm rather than parameter inflation.

**Memory Complexity:** Discrete models must cache intermediate hidden states across the entire sequence, resulting in a memory footprint of $\mathcal{O}(L \cdot d_h)$. In contrast, by employing the adjoint sensitivity method for backpropagation, the ODE framework reconstructs trajectories backward in time. This achieves a memory complexity of $\mathcal{O}(1)$ with respect to both $L$ and $N$, effectively preventing out-of-memory errors when processing long signals.

**Training Time:** The memory efficiency of the continuous formulation introduces a computational tradeoff during offline training. While discrete models execute $L$ deterministic state mappings with a time complexity of $\mathcal{O}(L \cdot d_h^2)$, the ODE performs $N$ sequential solver evaluations. Since capturing high-resolution physical dynamics typically requires $N > L$, the offline training complexity increases to $\mathcal{O}(N \cdot d_h^2)$.

**Inference Time:** During online execution, the ODE evaluates $\tilde{N}$ steps, yielding an inference time complexity of $\mathcal{O}(\tilde{N} \cdot d_h^2)$. Although theoretically governed by the number of solver steps rather than a fixed sequence length, this forward-only numerical integration remains deterministic and computationally lightweight, fully satisfying the real-time proactive handover requirements.

# C. Rationale for the Adaptive Clustering Mechanism

Real-world link failure detection operates in open-set environments where the number of degradation prototypes is unknown and site-specific. Consequently, standard methods requiring a pre-specified cluster count $K$ lack operational feasibility in blind deployments. As shown in Figure 7, forcing an arbitrary $K$ via K-means ($K \in \{2, 4, 6, 8, 10\}$) significantly increases FAR and FRR. Predefining $K$ imposes a rigid topological constraint, arbitrarily partitioning the latent space regardless of actual environmental complexity.

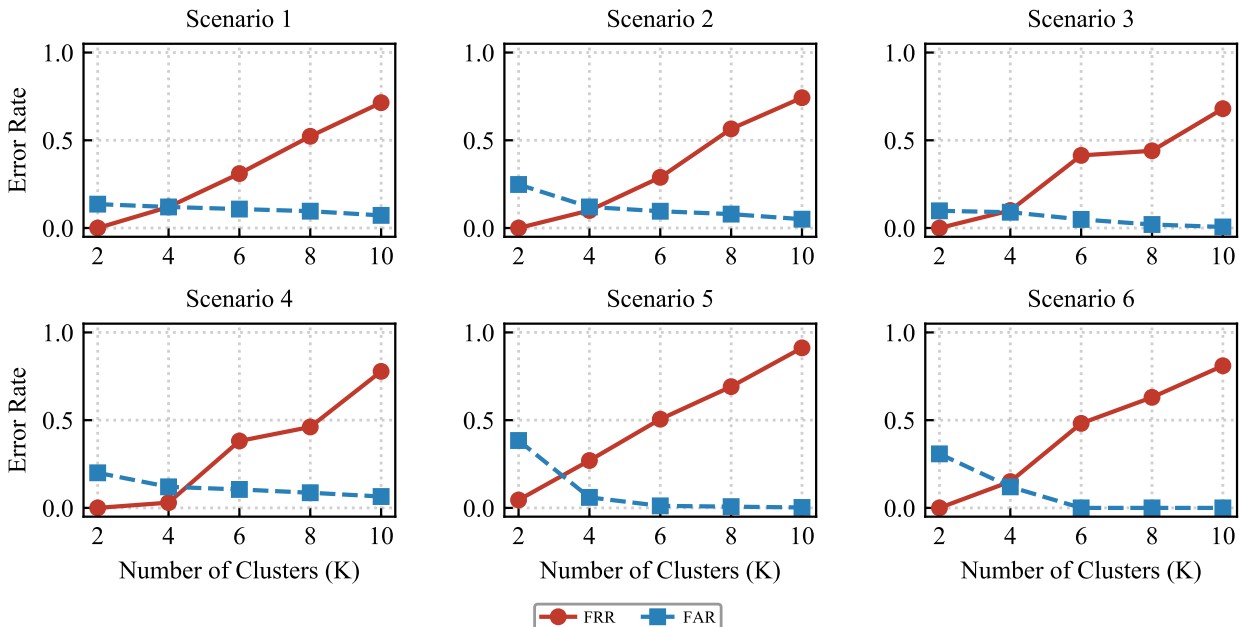

*Figure 7.* FAR and FRR against Number of Clusters for ODE.

Conversely, our adaptive threshold $\tau$ acts as a consistent physical resolution, bounding the maximum tolerable variance for intra-cluster trajectories. Once established, $\tau^*$ serves as an environment-adaptive ruler: naturally discovering fewer prototypes in simple scenarios and more in complex, multi-path ones. This paradigm shift from a fixed topological count ($K$) to a data-driven physical boundary ($\tau$) ensures robust deployment without manual retuning.

## D. Empirical Validation of the Continuous Prior

To validate the continuous prior and its impact on representation learning, we conducted a signal-level reconstruction test and a controlled clustering simulation.

### D.1. Signal-Level Validation of Continuous Dynamics

We evaluated the models' curve-fitting behaviors on synthetic 1D time series, comparing noisy smooth and discontinuous step signals (Figure 8). The results highlight contrasting inductive biases: the high-degree-of-freedom discrete model (ModernTCN) captures right-angle jumps but severely overfits injected noise. Conversely, our integration-constrained ODE acts as a stabilizer, achieving smooth denoising, albeit with slight hysteresis during abrupt jumps. This confirms the ODE's capacity to prioritize continuous physical evolution over transient fluctuations.

### D.2. Controlled Simulation for Adaptive Clustering

Building upon this stability, we designed a controlled simulation with a known ground-truth prototype count ($K = 4$) to evaluate unsupervised representation learning.

**Simulation Setup:** We generated 400 unlabeled sequences (100 per class) based on four physical trajectories: Linear Decay, Sinusoidal Fluctuation, Exponential Drop, and Step-wise Cliff. To simulate communication interference, these signals were obscured with Gaussian and high-frequency sinusoidal noise (Figure 9). All sequences underwent unsupervised reconstruction training, and their latent features were extracted for hierarchical clustering.

**Clustering Recovery:** This representational stability directly improves clustering accuracy. As shown in the dendrogram (Figure 10), the ODE latent space reveals a clear 4-class topological structure. By filtering out high-frequency jitter, our formulation accurately recovers both the true number of physical states ($K = 4$) and the continuous dynamics of the original

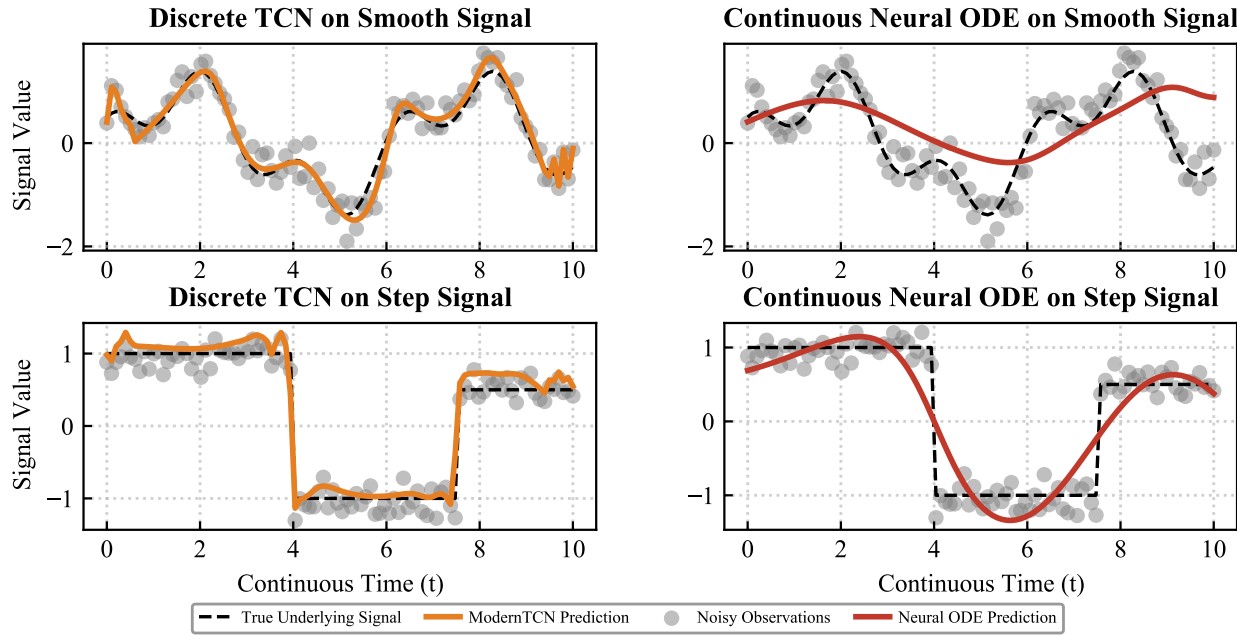

*Figure 8.* Reconstructing Smooth and Step Signals using ModernTCN and Neural ODE.

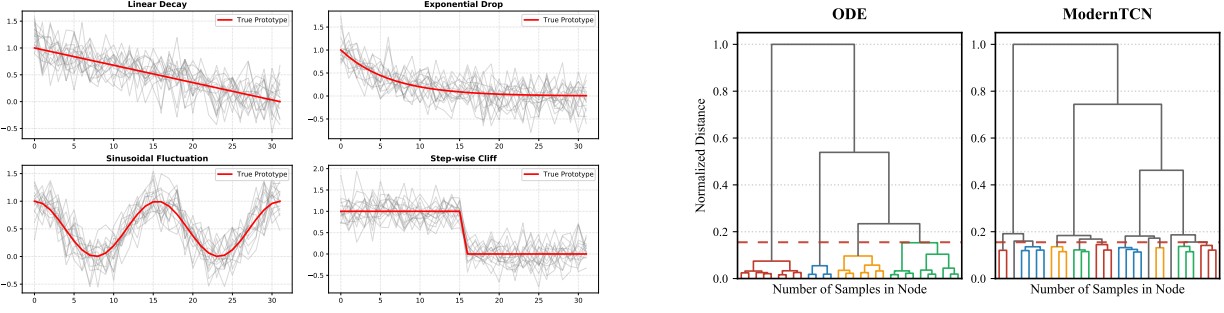

*Figure 9.* Test Data.

*Figure 10.* Hierarchical Structure of Prototypes for ODE and TCN.

prototypes (Figure 11). In contrast, the discrete architecture overfits the noise, erroneously fragmenting identical degradation trends into over ten overlapping micro-clusters (Figure 12).

# E. Assessment of Out-of-Distribution Behavior

To evaluate true out-of-distribution (OOD) performance under spatial relocation, we introduce a Cross-Environment Evaluation protocol centered on the concept of intent invariance. While the underlying movement intent (e.g., transitioning toward a target boundary) remains consistent, the spatial trajectories and radio environments change entirely, inducing a domain shift via unseen multipath signatures. We employ a rigorous zero-shot transfer protocol: both the model weights and the established degradation prototypes are derived exclusively from the source scenario. This represents a worst-case stress test, precluding any incremental learning or template calibration using target-environment data.

Table 3 quantifies the performance gap during this relocation, evaluated via cross-evaluations between distinct spatial scenarios (Scenario 2 and Scenario 3). We compare the intra-environment baseline (*Intra-Env*) against the zero-shot transfer (*OOD-Trans*). Discrete architectures experience a performance collapse (FRR > 0.90) upon relocation, as they strictly memorize environment-specific spatial fingerprints that fail to map to the new domain. Conversely, the continuous ODE maintains stable predictive performance. By capturing invariant physical dynamics, such as continuous velocity constraints,

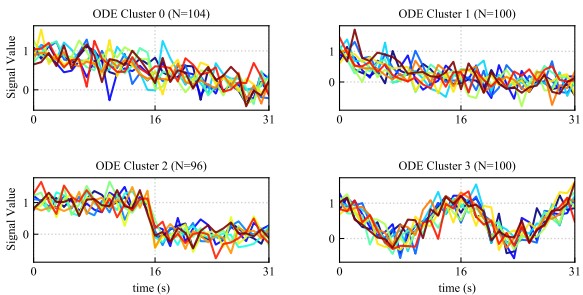

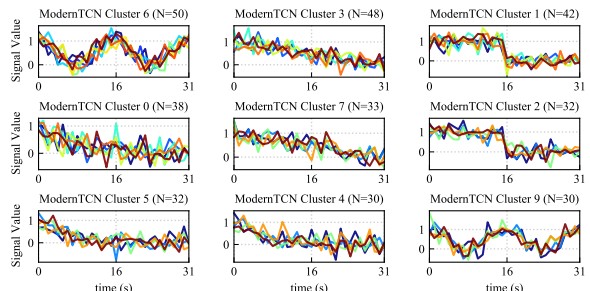

*Figure 11.* Visualization of Signal Series Prototypes for ODE.   *Figure 12.* Visualization of Signal Series Prototypes for TCN.

*Table 3.* OOD Generalization Performance: Evaluation of System Relocation Degradation.

| Method | Scenario 2 (Intra-Env) | | Scenario 2 (OOD-Trans) | | Scenario 3 (Intra-Env) | | Scenario 3 (OOD-Trans) | |
|---|---|---|---|---|---|---|---|---|
| | FAR | FRR | FAR | FRR | FAR | FRR | FAR | FRR |
| ResNet | $0.04 \pm 0.01$ | $0.66 \pm 0.15$ | $0.04 \pm 0.01$ | $0.96 \pm 0.04$ | $0.03 \pm 0.01$ | $0.93 \pm 0.06$ | $0.04 \pm 0.02$ | $0.94 \pm 0.05$ |
| LSTM | $0.36 \pm 0.09$ | $0.13 \pm 0.12$ | $0.13 \pm 0.08$ | $0.73 \pm 0.14$ | $0.14 \pm 0.07$ | $0.33 \pm 0.09$ | $0.28 \pm 0.09$ | $0.50 \pm 0.08$ |
| GRU | $0.03 \pm 0.02$ | $0.09 \pm 0.10$ | $0.04 \pm 0.01$ | $0.85 \pm 0.13$ | $0.01 \pm 0.02$ | $0.29 \pm 0.19$ | $0.04 \pm 0.01$ | $0.87 \pm 0.06$ |
| ModernTCN | $0.04 \pm 0.01$ | $0.54 \pm 0.21$ | $0.05 \pm 0.02$ | $0.93 \pm 0.06$ | $0.04 \pm 0.01$ | $0.61 \pm 0.12$ | $0.04 \pm 0.02$ | $0.95 \pm 0.04$ |
| ODE (Ours) | $0.04 \pm 0.02$ | $0.00 \pm 0.01$ | $0.05 \pm 0.03$ | $0.28 \pm 0.11$ | $0.03 \pm 0.01$ | $0.00 \pm 0.01$ | $0.05 \pm 0.04$ | $0.31 \pm 0.16$ |

the ODE generalizes beyond site-specific noise. The subsequent FRR increase observed during *OOD-Trans* reflects the inevitable spatial relocation degradation caused by latent distribution shifts from novel multipath profiles.

While the zero-shot setting provides a theoretical lower bound on robustness, practical deployments facilitate rapid adaptation. The embedded physical constraints enable the ODE framework to align with new scenarios using significantly fewer samples than standard discrete networks. Furthermore, the degradation templates can be efficiently updated with minimal target-environment observations, ensuring seamless operational carry-over without full model retraining.

# F. Extended Visualization of Learned Prototypes

In this appendix, we provide a comprehensive visualization of the learned representations across all physical scenarios. For each scenario and comparative method, we present two complementary views to validate the clustering quality:

- **Hierarchical Structure of Prototypes:** The dendrogram illustrating how the model groups latent dynamics and determines the cut-off threshold.

- **Visualization of Signal Series Prototypes:** The alignment between the reconstructed signal series and the extracted physical prototypes (highlighted as colored curves).

As illustrated in the subsequent figures, our method consistently demonstrates superior performance. By modeling the latent state evolution as a continuous integral curve, ODE effectively captures the underlying differential properties of the signal. This mechanism results in **compact, well-separated clusters**, where the learned prototypes accurately track the central dynamic trend of the degradation patterns.

In stark contrast, discrete baselines (e.g., GRU, LSTM) frequently struggle to distinguish complex dynamics. Their results often exhibit **sparse, fragmented clusters** or numerous "outlier" groups that lack physical interpretability. This limitation arises from their tendency to overfit high-frequency stochastic noise rather than recovering the smooth, continuous evolution of the physical process. Our code and data are provided in the supplementary material.

## F.1. Scenario 1

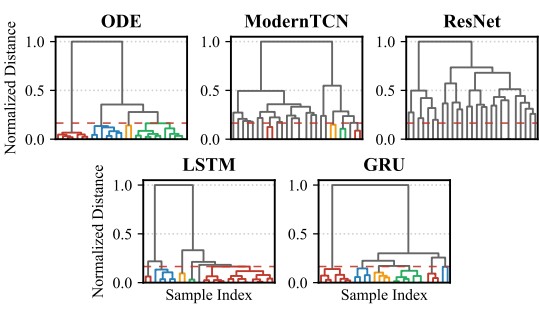

*Figure 13.* Hierarchical Structure of Prototypes. The dendrogram reveals the semantic grouping of latent dynamics, where the red dashed line denotes the **adaptive cut-off threshold**. Sub-trees merging below this threshold are identified as coherent physical prototypes.

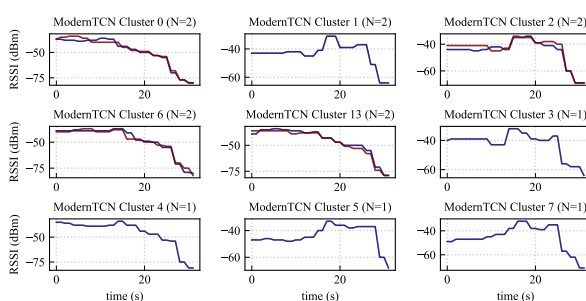

*Figure 14.* Visualization of Signal Series Prototypes for ModernTCN. Due to the excessive number of fragmented clusters, we visualize only the top-9 most significant categories (ranked by cluster size). The highlighted colored curves represent the corresponding prototypes, capturing the central dynamic trend of each category.

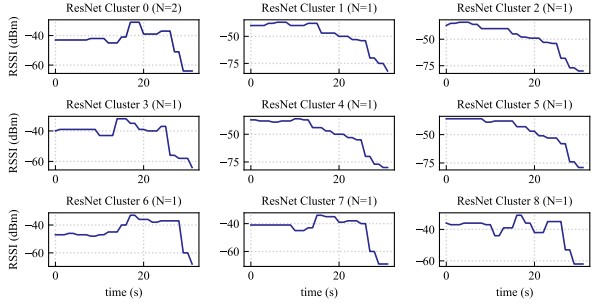

*Figure 15.* Visualization of Signal Series Prototypes for ResNet. Due to the excessive number of fragmented clusters, we visualize only the top-9 most significant categories (ranked by cluster size). The highlighted colored curves represent the corresponding prototypes, capturing the central dynamic trend of each category.

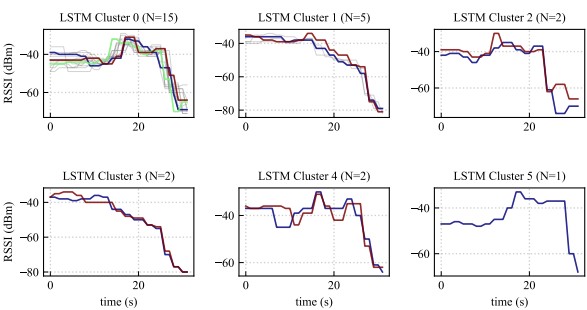

*Figure 16.* Visualization of Signal Series Prototypes for LSTM. The figure displays all the signal series assigned to each cluster. The highlighted colored curves represent the corresponding prototypes, capturing the central dynamic trend of each category.

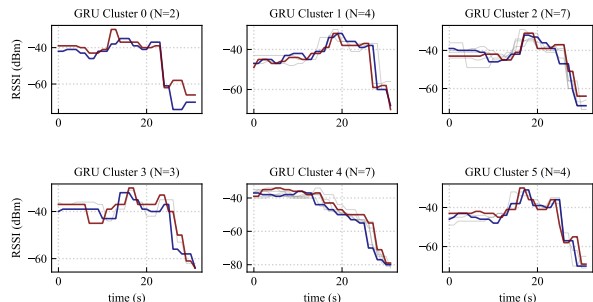

*Figure 17.* Visualization of Signal Series Prototypes for GRU. The figure displays all the signal series assigned to each cluster. The highlighted colored curves represent the corresponding prototypes, capturing the central dynamic trend of each category.

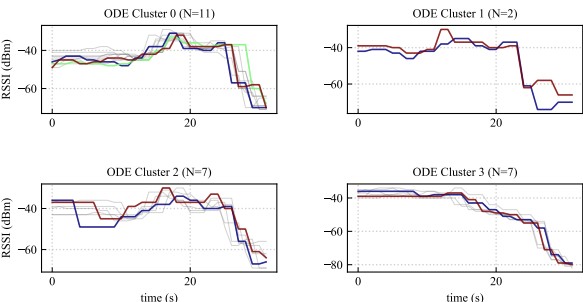

*Figure 18.* Visualization of Signal Series Prototypes for ODE. The figure displays all the signal series assigned to each cluster. The highlighted colored curves represent the corresponding prototypes, capturing the central dynamic trend of each category.

## F.2. Scenario 2

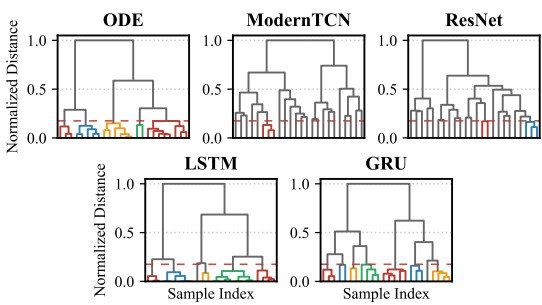

*Figure 19.* Hierarchical Structure of Prototypes.

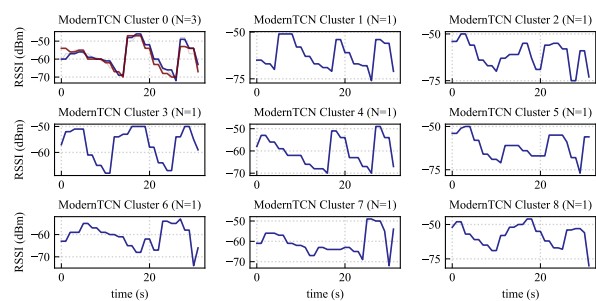

*Figure 20.* Visualization of Signal Series Prototypes for ModernTCN.

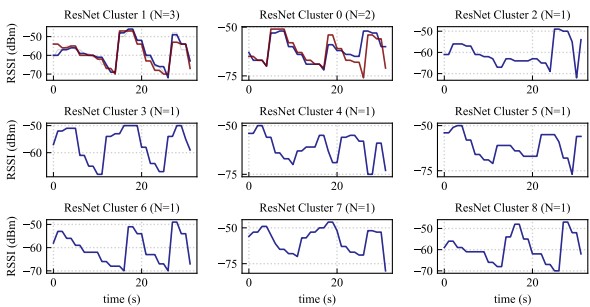

*Figure 21.* Visualization of Signal Series Prototypes for ResNet.

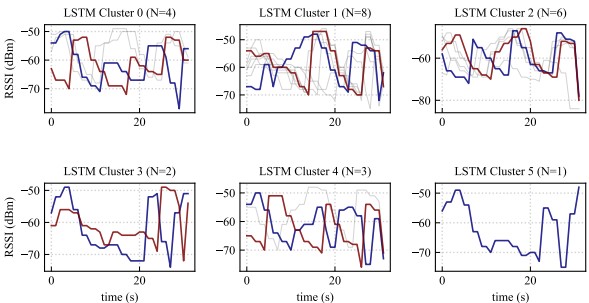

*Figure 22.* Visualization of Signal Series Prototypes for LSTM.

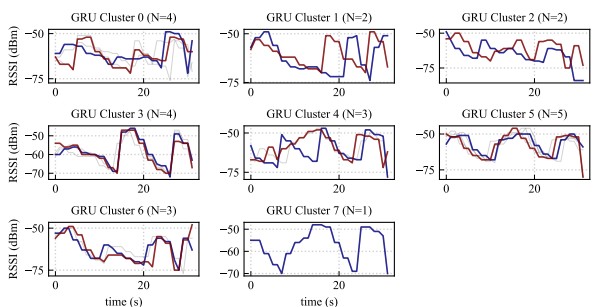

*Figure 23.* Visualization of Signal Series Prototypes for GRU.

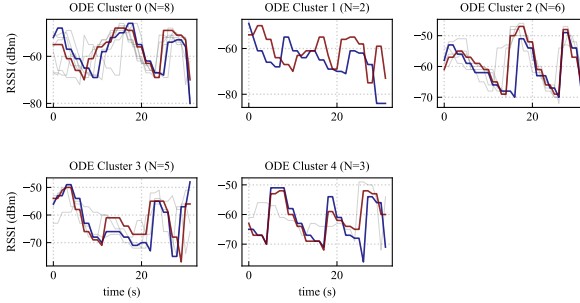

*Figure 24.* Visualization of Signal Series Prototypes for ODE.

## F.3. Scenario 3

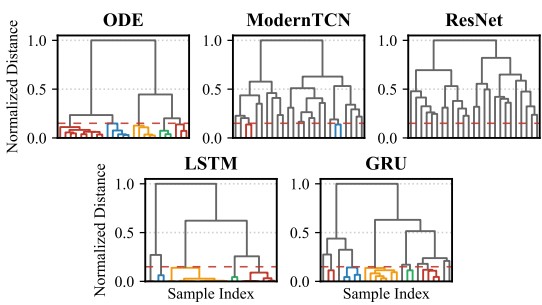

*Figure 25.* Hierarchical Structure of Prototypes.

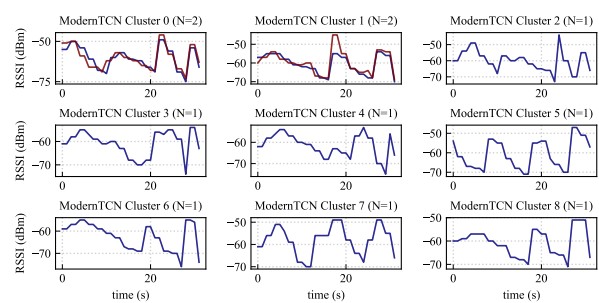

*Figure 26.* Visualization of Signal Series Prototypes for ModernTCN.

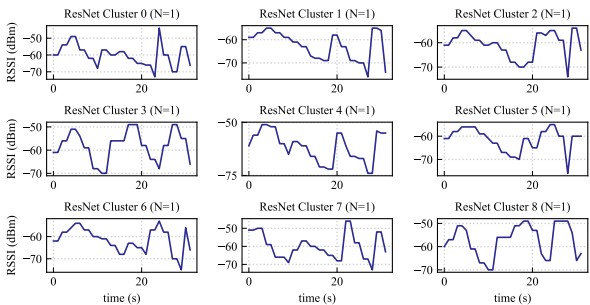

*Figure 27.* Visualization of Signal Series Prototypes for ResNet.

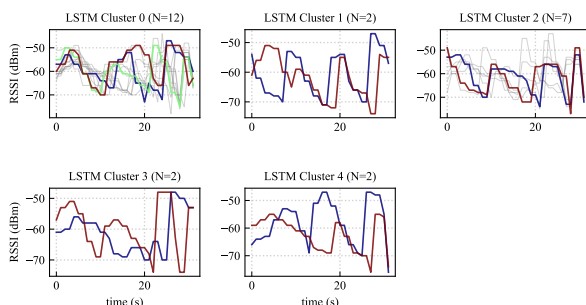

*Figure 28.* Visualization of Signal Series Prototypes for LSTM.

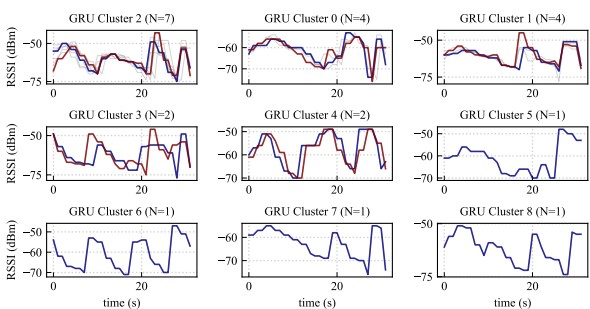

*Figure 29.* Visualization of Signal Series Prototypes for GRU.

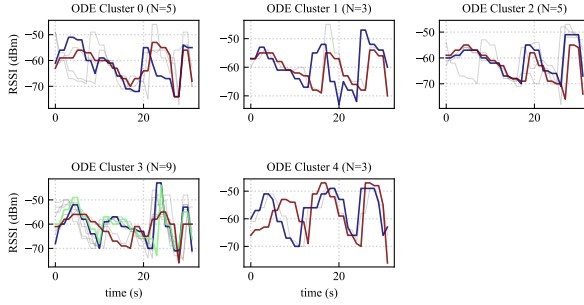

*Figure 30.* Visualization of Signal Series Prototypes for ODE.

## F.4. Scenario 4

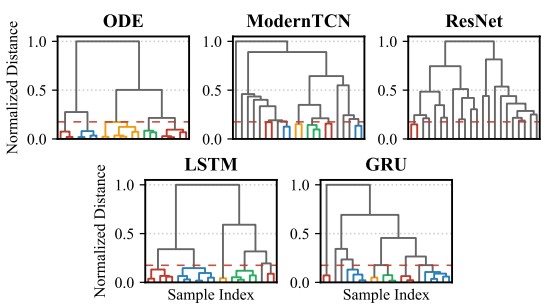

*Figure 31.* Hierarchical Structure of Prototypes.

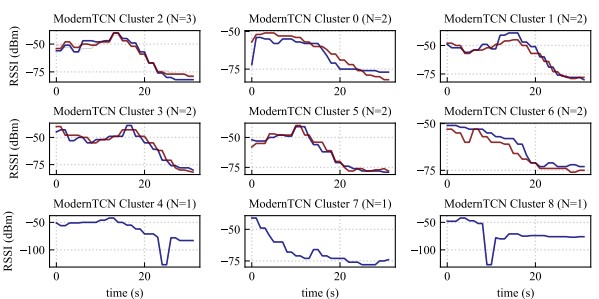

*Figure 32.* Visualization of Signal Series Prototypes for ModernTCN.

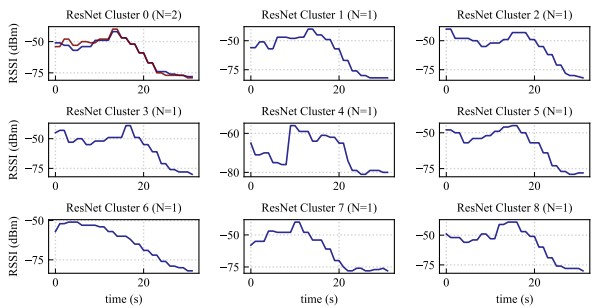

*Figure 33.* Visualization of Signal Series Prototypes for ResNet.

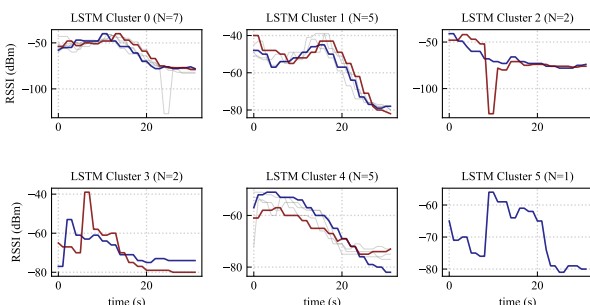

*Figure 34.* Visualization of Signal Series Prototypes for LSTM.

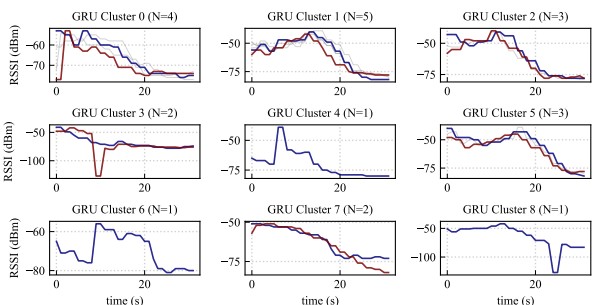

*Figure 35.* Visualization of Signal Series Prototypes for GRU.

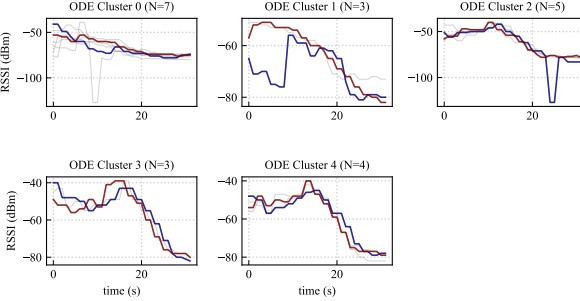

*Figure 36.* Visualization of Signal Series Prototypes for ODE.

## F.5. Scenario 5

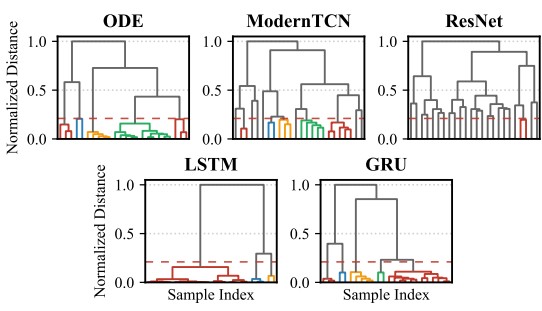

*Figure 37.* Hierarchical Structure of Prototypes.

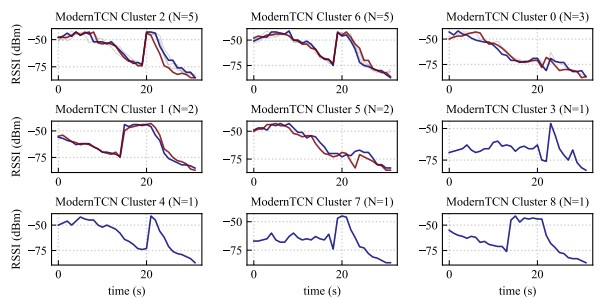

*Figure 38.* Visualization of Signal Series Prototypes for ModernTCN.

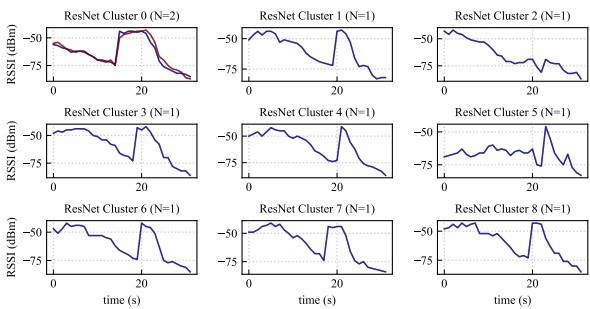

*Figure 39.* Visualization of Signal Series Prototypes for ResNet.

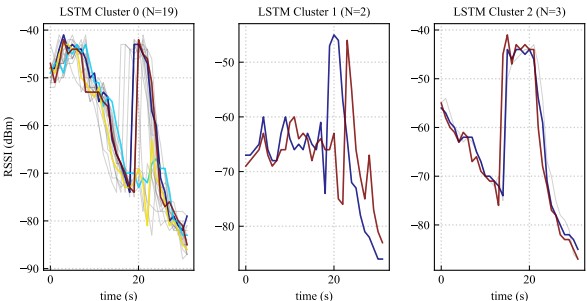

*Figure 40.* Visualization of Signal Series Prototypes for LSTM.

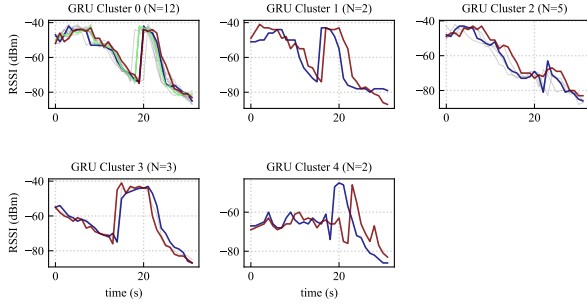

*Figure 41.* Visualization of Signal Series Prototypes for GRU.

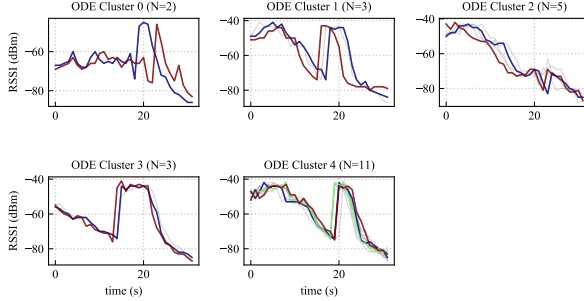

*Figure 42.* Visualization of Signal Series Prototypes for ODE.

## F.6. Scenario 6

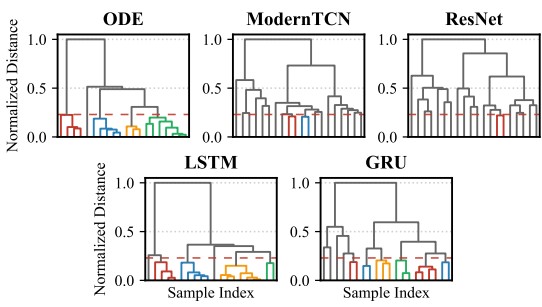

*Figure 43.* Hierarchical Structure of Prototypes.

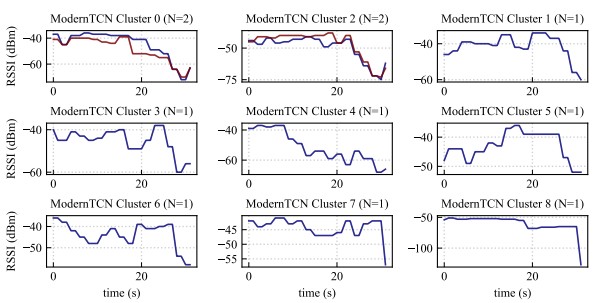

*Figure 44.* Visualization of Signal Series Prototypes for ModernTCN.

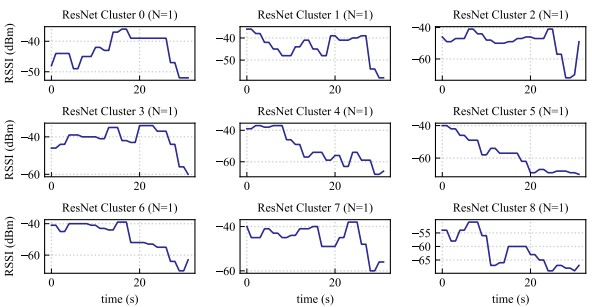

*Figure 45.* Visualization of Signal Series Prototypes for ResNet.

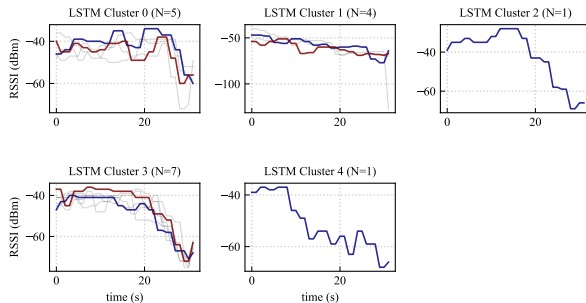

*Figure 46.* Visualization of Signal Series Prototypes for LSTM.

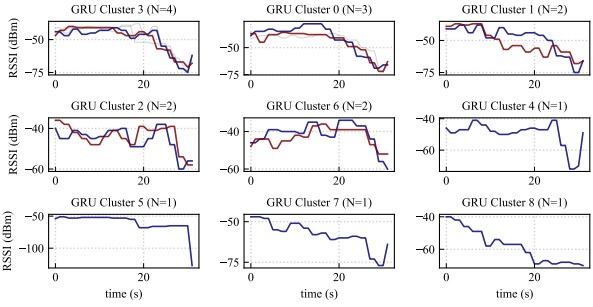

*Figure 47.* Visualization of Signal Series Prototypes for GRU.

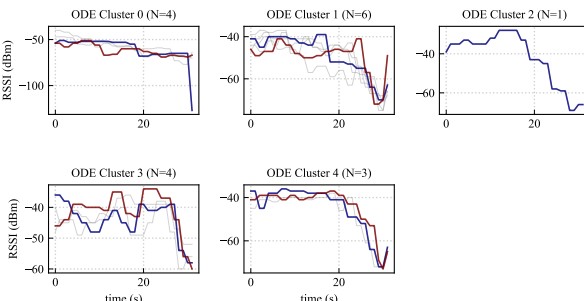

*Figure 48.* Visualization of Signal Series Prototypes for ODE.

