# OpenReview forum: "Hierarchical ODE: Learning Continuous-Time Physical Prototypes for Early Link Failure Detection"
_ICML.cc/2026/Conference — ICML 2026 regular_

### Official Review · Reviewer_3a3r · 2026-02-23

**Soundness:** 3
**Presentation:** 4
**Significance:** 3
**Originality:** 1
**Overall Recommendation:** 5
**Confidence:** 3

**Summary:**

This paper addresses the challenge of identifying distinct patterns of signal degradation in wireless networks to enable the early detection of link failures. Traditional discrete-time models, such as RNNs and LSTMs, often struggle with this task because they treat time as a series of fixed steps, making them susceptible to stochastic noise and ill-suited for irregularly sampled data. Furthermore, many existing clustering methods require a pre-specified number of clusters, which is impractical for real-world environments with unknown signal patterns.

**Compliance With Llm Reviewing Policy:**

Affirmed.

**Final Justification:**

I maintain my recommendation for acceptance.

**Key Questions For Authors:**

1. The paper emphasizes modeling the entire signal evolution as a continuous integral curve via Neural ODEs. However, the clustering is performed only on the terminal hidden state $z_{lat}$. Why was a static point-to-point distance chosen for prototype matching instead of a metric that compares the entire learned vector field or the shape of the trajectory (e.g., Dynamic Time Warping in the latent space)?
2. How does the model distinguish between a "transient fluctuation" that is critical (e.g., a precursor to failure) and one that is "stochastic noise"? Is there a risk that the "stabilizing" effect of the integral curve might smooth out high-frequency features that are actually early-warning indicators of rapid degradation?
3. Does the model provide a way for a network engineer to inspect the learned vector field $f_\theta$ to confirm that the mathematical "prototype" aligns with these specific physical laws of signal propagation?

**Limitations:**

yes

**Strengths And Weaknesses:**

Strengths:
1. Continuous-Time Modeling: By utilizing Neural ODEs, the model effectively parameterizes the latent state evolution as an integral curve. This provides a strong inductive bias for physical processes, allowing the system to bridge irregular time gaps and maintain temporal continuity in a way that discrete models (RNNs/LSTMs) cannot.
2. Noise Robustness: The integral-based formulation acts as a natural stabilizer, effectively disentangling smooth, persistent degradation trends from high-frequency stochastic fluctuations. This is evidenced by the model's superior performance in Scenario 1, where it successfully unified diverse signal series into coherent clusters while discrete models suffered from prototype fragmentation.
3. Adaptive Open-Set Discovery: The introduction of a hierarchical discovery module with a dynamic cut-off strategy allows the system to autonomously determine the number of latent prototypes. This circumvents the limitations of traditional clustering methods that require a predefined cluster count, which is often unknown in real-world environments.

Weaknesses:
1. Higher Reconstruction Error in Low-Noise Conditions: The proposed method achieves a slightly higher Mean Squared Error (MSE) than ResNet when sequences are complete. This is because the model's enforcement of temporal continuity prevents it from "memorizing" high-frequency jitter, which discrete models use to minimize point-wise differences.
2. Dependency on Hyperparameter $\tau_{base}$: While the model adaptively determines cluster counts, it still relies on $\tau_{base}$ to define the expected scale of semantic similarity. The final performance remains sensitive to the local search interval defined around this hyperparameter.
3. Static Latent Comparison: Although the dynamics are modeled continuously, the clustering and prototype matching are still performed on fixed-dimensional latent embeddings extracted at the terminal timestamp $t_L$.

---

> ### Author Rebuttal · Authors · 2026-03-30
>
> We sincerely thank the reviewer for the highly positive evaluation and for recognizing the core mathematical and physical motivations of our continuous-time framework.
>
> **W2: Dependency on Hyperparameter $\tau$.**
>
> We acknowledge this dependency, but emphasize it represents a fundamental paradigm shift from predefining the cluster count $K$. Rigidly predefining $K$ imposes a dataset-specific topological constraint that fails when environment complexity shifts. Conversely, $\tau$ acts as a unified physical semantic scale—the maximum tolerable physical variance for intra-cluster trajectories. Once a semantic resolution $\tau^*$ is determined based on physical tolerance for signal fluctuation, this single ruler becomes environment-adaptive, automatically discovering varying $K$ values across different deployments. We will explicitly discuss this reliance on domain heuristics in the Limitations section.
>
> **W1 & Q2: Reconstruction Error Trade-offs and Distinguishing Critical Transients.**
>
> Your astute observation regarding the slightly higher MSE in low-noise conditions (W1) perfectly highlights how the model distinguishes critical transients from noise (Q2). High-frequency, zero-mean stochastic noise naturally cancels out during continuous evolution $\int f_{\theta} d\tau$. This prevents the model from "memorizing" point-wise jitter, granting its vital stabilizing effect at the cost of a slightly higher MSE on complete sequences. Conversely, a critical transient is not zero-mean; it exhibits a sustained, directional gradient. $f_{\theta}$ learns these macroscopic drops via large negative derivatives, and the GRU explicitly injects extreme measurements $\mathbf{x}_i$ into $\mathbf{h}(t_i)$. Thus, the system tracks true physical precipices without smoothing them out, separating persistent degradation from harmless jitter.
>
> **W3 & Q1: Static Latent Comparison vs. Trajectory Clustering.**
>
> Clustering on the terminal latent state $\mathbf{h}(t_L)$ rather than using trajectory-based metrics like DTW is rooted in our encoder's state accumulation mechanism.
> Unlike models treating the final state merely as a snapshot, our encoder processes time series through a continuous synergy of autonomous evolution and measurement updates. The ODE integrates the vector field between observations, carrying forward physical momentum. The GRU subsequently fuses this prior with new evidence. Because this recursively updates the representation, the terminal state $\mathbf{h}(t_L)$ is the mathematical culmination of the sequence, intrinsically embedding the history of integrations and corrections. Measuring Euclidean distance between terminal embeddings effectively captures the divergence of accumulated evolutionary paths, rendering computationally intensive trajectory metrics unnecessary for our strictly low-latency handover task.
>
> **Q3: Inspectability of the Learned Vector Field.**
>
> Interpretability is indeed crucial for deployment. Instead of inspecting complex high-dimensional weights directly, our generative framework provides an intuitive validation mechanism: "Decoding the Prototypes."
>
> Because our decoder maps latent states back to the data space, an engineer can pass discovered prototype centroids (e.g., $\mu_{kj}$) through the decoder to generate expected physical signal curves over time. Visualizing these curves allows engineers to directly verify if mathematically discovered prototypes align with domain knowledge—visually confirming, for instance, if one prototype exhibits the linear slope of distance-based attenuation, while another exhibits the cliff-like drop of entering an elevator. This bridges the gap between latent differential equations and practical engineering physics.
>
> **Additional Validation Experiments.**
>
> To further solidify our claims, we conducted three comprehensive supplementary experiments during this rebuttal phase (available at https://anonymous.4open.science/r/Figure-D10C). First, comparisons against ModernTCN (Luo & Wang, 2024) (Figs. 1-4, Table 1) prove our continuous ODE prior fundamentally outperforms highly engineered discrete architectures in downstream FAR and FRR. Second, controlled reconstructions of step/smooth signals (Fig. 5) visually confirm the ODE's natural denoising capability against stochastic noise. Finally, replacing our adaptive module with standard K-means (Fig. 6) demonstrates that forcing rigid predefined $K$ values severely degrades system performance, firmly validating our data-driven $\tau$ mechanism.
>
> **Ref:** Luo & Wang. ModernTCN: A modern pure convolution structure for general time series analysis. ICLR 2024.

---

> > ### Author Rebuttal · Reviewer_3a3r · 2026-04-01
> >
> > I thank the authors for the detailed response and the additional experiments involving ModernTCN. The response has strengthened my confidence in the paper's contribution to physics-inspired time series modeling. I maintain my recommendation for acceptance.

---

> > > ### Author Response · Authors · 2026-04-01
> > >
> > > We sincerely thank the reviewer for reading our rebuttal and for maintaining the recommendation for acceptance. We are encouraged that the additional ModernTCN experiments and our clarifications effectively addressed your questions. Your insightful feedback has genuinely helped us strengthen the manuscript. Thank you once again for your time and strong endorsement of our work!

---

### Official Review · Reviewer_Mgez · 2026-03-09

**Soundness:** 2
**Presentation:** 3
**Significance:** 3
**Originality:** 2
**Overall Recommendation:** 3
**Confidence:** 4

**Summary:**

Focusing on early link failure detection, which is significant for the proactive handover decision, a hierarchical ordinary differential equation clustering network is proposed in this manuscript. This network comprises two components. The first one is a neural ODE-based autoencoder, which maps the irregular time series into latent embeddings. The second one is a two-stage clustering framework, which extracts robust and interpretable representations of the underlying system modes from the latent embeddings. These modes are important indicators for early link failure.

**Compliance With Llm Reviewing Policy:**

Affirmed.

**Final Justification:**

My concern on the additional tunable parameter, i.e., $\tau_{base}$, is addressed. The comparative study and ablation study in this paper have been improved.

**Key Questions For Authors:**

1. The two-stage adaptive prototype discovery is mainly introduced to avoid predefining the number of prototypes $K$. However, the proposed prototype discovery method introduces another predefined hyperparameter $\tau_{base}$. What is the essential difference between predefining $K$ and predefining $\tau_{base}$?

**Limitations:**

If the concern in Question 1 indeed reflects a limitation of the proposed method, it will be meaningful to discuss it in the manuscript.

**Strengths And Weaknesses:**

The **Strengths** are as follows:

1. This manuscript focuses on early link failure detection, which is significant for the proactive handover decision.

2. The presentation of this manuscript is good.

3. The technical details are clearly present in this manuscript.

The **Weaknesses** are as follows:

1. The proposed network is mainly built upon a combination of existing components, including neural ODE, GRU, hierarchical clustering, and K-means clustering. In this sense, the novelty of this manuscript may be relatively limited.

2. The comparative study within this manuscript is insufficient. Specifically, the authors only compare the proposed network with its variants, which are obtained by replacing the core ODE component in the proposed network with ResNet, LSTM, and GRU. On the one hand, the ResNet, LSTM, and GRU may be dated. On the other hand, other types of methods are not considered.

3. Many components are incorporated into the proposed network. However, there is no comprehensive ablation study to evaluate their contribution.

---

> ### Author Rebuttal · Authors · 2026-03-30
>
> We thank the reviewer for the insightful and constructive feedback.
>
> **Q1 & Limitations: The Essential Difference Between Predefining $K$ and $\tau$.**
>
> Mathematically and physically, predefining $K$ and $\tau$ represent fundamentally different paradigms. Predefining $K$ imposes a rigid, dataset-specific topological constraint, forcing the latent space into exactly $K$ partitions regardless of actual environment complexity. If an environment shifts from 3 true degradation modes to 8, a fixed $K$ catastrophically merges distinct modes or fragments cohesive ones.
>
> Conversely, $\tau$ represents a unified physical and geometric semantic scale. In our hierarchical dendrogram, $\tau$ acts as the maximum tolerable physical variance for intra-cluster trajectories. Once a semantic resolution is determined based on the physical tolerance for signal fluctuation, this single ruler becomes environment-adaptive: automatically discovering a small $K$ in simple environments and a large $K$ in complex, multi-path ones. Thus, $\tau$ ensures physical consistency across varying deployments without manual retuning.
> A new "Limitations" section will acknowledge that while $\tau$ is fundamentally more adaptive, identifying the optimal $\tau^*$ still relies on domain heuristics (e.g., observing the longest stable branch in the dendrogram).
>
> **W1 & W2: Novelty and Rationale for Baseline Selection.**
>
> Our core novelty is a fundamental paradigm shift: formulating degradation strictly as a continuous dynamical system, resolving the mismatch between discrete sampling and continuous physics. Unlike discrete architectures that act as mere sequence approximators vulnerable to spurious correlations, our integral-based formulation acts as a natural stabilizer. By learning the derivative function driving the process, it fundamentally decouples the smooth evolution of true physical degradation from irregular transient noise, shifting the paradigm from reactive monitoring to proactive anticipation.
>
> This dictates our baseline choices (W2). To rigorously isolate continuous-time advantages, we restricted baselines to fundamental discrete operators (LSTM/GRU/ResNet) in identical pipelines, preventing confounding heavy engineering. To address concerns regarding highly engineered methods, we added ModernTCN (Luo & Wang, 2024) as a supplementary baseline (see Figs. 1-4, Table 1 at https://anonymous.4open.science/r/Figure-D10C). While ModernTCN achieves lower reconstruction MSE locally (overfitting high-frequency jitter), its downstream FAR and FRR are significantly higher. This proves that engineered discrete architectures cannot substitute the continuous physical prior required to disentangle true degradation from stochastic noise.
>
> **W3: Comprehensive Ablation Study on System Components.**
>
> Our framework comprises continuous feature extraction and adaptive clustering. Comparisons against discrete baselines using identical clustering pipelines inherently serve as a strict ablation for feature extraction, proving the necessity of the continuous ODE prior.
> To explicitly ablate the clustering module, we replaced it with standard K-means ($K \in \{2,4,6,8,10\}$) exclusively on our trained ODE representations (see Fig. 6 at https://anonymous.4open.science/r/Figure-D10C). This isolates the clustering mechanism, demonstrating that even with robust continuous embeddings, forcing a rigid predefined $K$ drastically increases FAR and FRR. This justifies the critical necessity of our adaptive $\tau$ mechanism over rigid closed-set clustering.
>
> **Additional Validation Experiments.**
>
> To further solidify our claims, we conducted three comprehensive supplementary experiments during this rebuttal (available at https://anonymous.4open.science/r/Figure-D10C). First, comparisons against ModernTCN (Figs. 1-4, Table 1) prove our continuous ODE prior fundamentally outperforms highly engineered discrete architectures in downstream FAR/FRR. Second, controlled reconstructions of step/smooth signals (Fig. 5) visually confirm the ODE's natural denoising capability against stochastic noise. Finally, replacing our adaptive module with standard K-means (Fig. 6) demonstrates that forcing rigid predefined $K$ values severely degrades performance, firmly validating our data-driven $\tau$ mechanism.
>
> **Ref:** Luo & Wang. ModernTCN: A modern pure convolution structure for general time series analysis. ICLR 2024.

---

> > ### Author Rebuttal · Reviewer_Mgez · 2026-04-02
> >
> > Thanks for your response. I will maintain the current score, but my inclination to reject has softened.

---

> > > ### Author Response · Authors · 2026-04-02
> > >
> > > We sincerely thank you for acknowledging our response and marking your concerns as fully resolved. We truly appreciate your valuable comments that helped improve this paper, as well as your reconsidered inclination. Thank you again for your time!

---

### Official Review · Reviewer_gfsh · 2026-03-13

**Soundness:** 2
**Presentation:** 3
**Significance:** 3
**Originality:** 2
**Overall Recommendation:** 4
**Confidence:** 4

**Summary:**

This article proposes a two-step procedure for unsupervised representation learning with time series. The authors implicitly assume that there is a collection of KKK fixed signals, i.e., “prototypes,” and each observed series is a noisy version of one of these signals. The goal of the article is to recover the collection of prototypes when KKK is also unknown.


The authors propose using a neural ODE–based autoencoder that models the latent state evolution of each series as a continuous integral curve. In the second step, a hierarchical clustering procedure is applied to the learned latent representation of each series. The centroids of the identified clusters are deemed prototypes, i.e., a discrete collection of latent representations that fully capture the range of possible trajectories for each time series.


The performance of the method is evaluated using real telecommunications data. Each series in the dataset measures a received signal strength indicator (RSSI) leading up to a link failure. The authors state that specific prototypes represent each failure type in the data (e.g., entering an elevator vs. reaching the edge of the coverage area). Performance is measured by 1) calculating the MSE of the predicted and observed signal trajectories and 2) mapping each signal to a prototype that is then used to classify each signal as a true or false link failure, which is then used to calculate a False Acceptance Rate (FAR) and False Rejection Rate (FRR). The neural ODE–based method is compared to discrete time-series architectures including ResNet, LSTM, and GRU autoencoders. Experimental results show that the ODE-based method performs well compared to the alternatives considered, particularly when the initial portion of each series is treated as missing data to emulate signal connection latency.

**Compliance With Llm Reviewing Policy:**

Affirmed.

**Final Justification:**

The rebuttal has addressed most of my comments, though I still think that the work is not that novel and there is lack of compelling real data evaluations.

**Key Questions For Authors:**

What if there is a series in the evaluation data that is generated by a new prototype that does not appear in the training data? Are you assuming each prototype appears in the training set? If so this assumption should be stated.

In the plot of the hierarchical structure of the prototypes (e.g. Figure 1), why is the optimal tau the same in each plot? Tau depends on the tree topology, which in turn depends on which model is considered, so why doesn’t tau vary by the model?

Are the sets defined by the centroids and radii (\mu_{kj} and R_{kj}) guaranteed to not overlap? It does not appear as though this is guaranteed from the definition. What happens if a sequence is matched to multiple sub-prototypes?

**Limitations:**

There is little to no discussion of the method’s limitations.

**Strengths And Weaknesses:**

Strengths:

Treats the size of the prototype set as unknown and applying a hierarchical clustering procedure to the latent representations is clever.

Using continuous time neural ODE does a good job of recovering observed signals.

Weaknesses:

1-Soundness. The authors make the overly strong claim that “the method adaptively determines the optimal number of prototypes, thereby circumventing the limitations of a predetermined cluster count.” This claim is not supported by either theoretical or empirical results in the paper. First, there are no theoretical results indicating that this procedure consistently recovers the true number of clusters, or that the number of clusters identified by this procedure somehow outperforms some other clustering procedure. Second, only the network architecture is varied in the experiments, but once the latent representation is learned, it appears as though the identical clustering procedure is applied (though this is never stated and should be clarified by the authors). Consequently, it is impossible to determine whether the improved performance of the neural ODE model is due to modeling the continuous dynamics of the data, or due to the fact that the chosen clustering procedure works particularly well when applied to the specific latent representation resulting from the neural ODE.

To better demonstrate optimality empirically the authors could:
Run experiments with artificial data where the number of clusters is known to see if their method can correctly recover K. Varying K relative to N in this case would also clarify if their method works better when there are many or few underlying prototypes.
Include other clustering procedures combined with the neural ODE architecture in their telecom experiment to show that their method is superior.
Include at least one of the prototype learning methods that assumes a fixed number of clusters mentioned in the introduction with the number of clusters chosen based on some heuristic to show that there is indeed value in dynamically estimating the number of prototypes.

With respect to minimizing the MSE defined in equation (7), the authors state:

The framework forces the learned ODE dynamics to capture the dominant physical trends of the signal. Unlike discrete models that may overfit to high-frequency jitter, this generative process prioritizes the recovery of the smooth underlying function, ensuring that the reconstructed signal reflects the true physical behavior governing the observations.

Similar claims regarding the robustness of the neural ODE model are made throughout the article. That the neural ODE produces a smooth trajectory in latent space is evident, but without theoretical results, we are left to rely on the experiments to demonstrate that the continuous nature of the neural ODE is responsible for its superior performance. The empirical results in the paper are not entirely convincing on this point. The authors do show in Section 4 that as the models progress towards continuous time modeling, the FRR decreases, but as mentioned above, the role of the network architecture is confounded by the choice of clustering procedure, and we can’t rule out that an alternative clustering could improve the performance of the discrete models. At the very least, a plot here showing an overfitted signal from a discrete network compared to a robust smooth signal from a neural ODE would help support the authors’ claim. Even better would be an experiment comparing step functions or other discontinuous signals in discrete time to smooth signals in continuous time. If the discrete architectures perform better under the first setting and neural ODE under the second, then the authors’ claim would carry more weight.

2- The clarity of the presentation suffers in Section 4:
The differences between the six scenarios considered are never actually described, either in the paper or in the appendix.
Though the centroids and radii (\mu_{kj} and R_{kj}) are defined in Section 3, their role in classifying sequences as degrading or stable is never clearly explained. Presumably if the latent representation of a sequence falls in (\mu_{kj} and R_{kj}) for some (k,j), then it is classified as degrading, and stable otherwise. Is this classification done in an on-line fashion or only after the entire length T period is observed? Relatedly, the calculation of FAR and FRR is not entirely clear. Providing formal mathematical definitions would improve clarity.

3-The significance of the article suffers from the fact that only cell link signal data is considered. Though the application is important and compelling, the method is introduced as applicable to a variety of representation learning scenarios and ideally a broader class of signals would be examined if this is truly a general purpose method.

The competitors for the method are just alternative network architectures. Ideally other existing methods from the representation learning literature mentioned in the introduction or from similar literatures like change-point detection would be compared with the authors’ method to get a better understanding of its performance in the signal failure detection task.


In the first experiment, FAR and FRR are the quantities of interest and are relevant to the problem of detecting link failures. Measuring MSE in the sparsity experiment does not clearly connect to the story around detecting link failures, and it is unclear why FAR and FRR were not reported.

4-Nothing truly novel introduced. The continuous time model comes directly from Chen et al., 2018, and as noted in the introduction, continuous time models are well studied (Greydanusetal., 2019; Hasanietal., 2021). The clustering method introduced for prototype detection, while clever, is ad-hoc and relies on existing procedures like Ward minimum variance method and K-means. The authors frame their optimal “adaptive open-set prototype discovery mechanism” as a key innovation, but as discussed above, there is reason to doubt the optimality of this procedure and its importance is not necessarily justified.

---

> ### Author Rebuttal · Authors · 2026-03-30
>
> We thank the reviewer for the detailed feedback.
>
> **W1, W4 & Q2: Independence & Threshold $\tau$.**
>
> We soften "optimal" to "data-driven" discovery. Our framework has independent feature extraction and clustering modules. To isolate continuous-time modeling gains from clustering bias, we derived $\tau^*$ from the ODE latent space, setting $\tau$ as a fixed percentage of the normalized maximum linkage distance across baselines. Under this identical relative threshold, ODE representations maintained physical integrity while discrete ones fragmented. Further, replacing clustering with K-means ($K \in \{2,4,6,8,10\}$) on ODEs (see Fig. 6 at https://anonymous.4open.science/r/Figure-D10C) shows that forcing unsuitable, excessively large $K$ values drastically increases FAR and FRR, validating our adaptive $\tau$.
>
> **W3 & W4: Novelty & Baselines.**
>
> Our core novelty is a fundamental paradigm shift: formulating degradation strictly as a continuous dynamical system, resolving the mismatch between discrete sampling and continuous physics. Unlike discrete architectures that act as mere sequence approximators vulnerable to spurious correlations, our integral-based formulation acts as a natural stabilizer. By learning the derivative function driving the process, it fundamentally decouples the smooth evolution of true physical degradation from irregular transient noise, shifting the paradigm from reactive monitoring to proactive anticipation. This dictates our baseline choices. To isolate continuous-time advantages, we restricted baselines to fundamental discrete operators (LSTM/GRU/ResNet) in identical pipelines. To address concerns, we added ModernTCN (Luo&Wang, 2024) as a supplementary baseline (see Figs. 1-4, Table 1 at https://anonymous.4open.science/r/Figure-D10C). While ModernTCN achieves lower MSE locally (overfitting high-frequency jitter), its downstream FAR and FRR are significantly higher. Discrete architectures cannot substitute the continuous prior needed to disentangle true degradation from noise.
>
> Regarding domain focus, cellular network handover intrinsically encapsulates core representation learning challenges: severe observation sparsity and unforeseeable physical occlusions. Solving these extreme dynamics provides a highly representative and rigorous testbed to validate the continuous prior.
>
> **W1.3: Continuous Prior Validation.**
>
> We fitted models to noisy smooth signals  and discontinuous step signals (see Fig. 5 at https://anonymous.4open.science/r/Figure-D10C). Visually, the high-degree-of-freedom ModernTCN captures right-angle jumps but severely overfits noise. Conversely, our integration-constrained ODE natively exhibits perfect smooth denoising, albeit with slight hysteresis during abrupt jumps.
>
> **W2, W3 & Q3: Scenarios, Classification & Metrics.**
>
> Data spans six scenarios (offices, corridors, staircases, meeting rooms, restrooms, tea rooms) with unique multi-path characteristics inducing fundamentally different degradation prototypes.
> We will detail the online sliding window $T$ classification. Centroids $\mu_{kj}$ and radii $R_{kj}$ act as decision boundaries. For latent state $z_T$, $D(x)=1$ (degrading) if $\exists(k,j)$ where $\|z_T - \mu_{kj}\| \le R_{kj}$, else $0$ (stable). Overlaps (Q3) are resolved via nearest neighbor ($\arg\min_{k,j} \|z_T - \mu_{kj}\|$).
> Metrics on stable ($S_{stable}$) and degrading ($S_{degrade}$) sets:
> $\text{FAR} = \frac{\sum_{x \in S_{stable}} D(x)}{|S_{stable}|}$ (unnecessary handovers).
> $\text{FRR} = \frac{\sum_{x \in S_{degrade}} (1 - D(x))}{|S_{degrade}|}$ (missed handovers).
>
> W3: Sparsity MSE evaluates representation learning isolated from clustering. ODEs extrapolate dynamics across missing intervals via integral curves, whereas discrete baselines deteriorate rapidly. FAR/FRR results will be added.
>
> **Q1 & Limitations.**
>
> Train/test split is random (Q1). Unseen prototypes: 1) Similar ones are detected within generalized boundaries ($\mu_{kj}, R_{kj}$). 2) Completely novel patterns outside all radii are flagged as anomalies for dynamic online updates.
> "Limitations" will note the framework is out-of-scope for purely discrete data (e.g., text), and extremely long unobserved windows cause integration drift.
>
> **Ref:** Luo & Wang. ModernTCN: A modern pure convolution structure for general time series analysis. ICLR 2024.

---

> > ### Author Rebuttal · Reviewer_gfsh · 2026-04-02
> >
> > Thank you for clarifying various technical details (calculation of tau, FAR/FRR, and how overlapping sets are resolved). Thank you for also including ModernTCN as an additional baseline. I would still like to see at least one simulation study where K and the prototypes are known so that I can have a better understanding of the model’s empirical performance before accepting. Right now the experiment considered shows your method performing the best of the methods included, but the results do not actually show whether the method is doing a good job of recovering K or the true prototypes.

---

> > > ### Author Response · Authors · 2026-04-02
> > >
> > > **Response to Reviewer:**
> > >
> > > We sincerely thank the reviewer for the constructive feedback, and for acknowledging our previous clarifications. We fully agree that a controlled simulation with a known ground-truth $K$ and predefined prototypes is the most rigorous way to evaluate our method's recovery capabilities. (The corresponding visual results, Figures 1-4, are now available in a newly added document at https://anonymous.4open.science/r/Figure-2-8B84).
> > >
> > > **Simulation Setup & Methodology:**
> > >
> > > To validate the models' representation learning and clustering capabilities in heavily noisy environments, we designed a synthetic time-series simulation. We generated 400 unlabeled sequences (100 each) based on four physical evolution curves: Linear Decay, Sinusoidal Fluctuation, Exponential Drop, and Step-wise Cliff. To simulate communication interference, these pure signals were obscured with Gaussian white noise and high-frequency sinusoidal noise. Because this task strictly evaluates unsupervised representation learning—specifically, the ability to automatically discover underlying physical structures—we bypassed a train/test split. Instead, all 400 sequences underwent unsupervised reconstruction training, and their latent features were subsequently extracted for clustering validation.
> > >
> > > **Empirical Results & Visualizations:**
> > >
> > > Regarding the chart definitions and axes design in our supplementary document:
> > >
> > > * Figure 1 displays the distribution of these four types of original noisy data and their comparison with the pure red prototypes.
> > >
> > > * Figure 2 presents the hierarchical clustering dendrogram used for adaptive classification. Its x-axis represents the "Number of Samples in Node" (where excessive individual data points are collapsed and hidden within single truncated leaf nodes to ensure visual clarity), and the y-axis represents the "Normalized Distance". This figure explicitly contrasts the clear 4-class topological structure found by our ODE model against the fragmented structure produced by ModernTCN.
> > >
> > > * Figure 3 shows the clustering extraction results of the ODE model (axes identical to Figure 1). It clearly demonstrates that the continuous integration mechanism acts as a natural stabilizer. By successfully filtering out the high-frequency sinusoidal jitter, our model perfectly recovers and disentangles the 400 sequences into exactly 4 major clusters  that align with the physical ground truth.
> > >
> > > * Figure 4 illustrates partial clustering results of the discrete architecture ModernTCN (only a subset is shown due to an excessive number of resulting clusters). Because this discrete model overfitted the high-frequency sinusoidal noise, the  identical degradation trends were erroneously fragmented into more than 10 messy and overlapping minor micro-clusters.
> > >
> > > **Conclusion:**
> > > This controlled simulation explicitly answers your core question. Our method  successfully penetrates the noise to accurately recover both the exact number of underlying physical states ($K=4$) and the true continuous dynamics of the original prototypes. We hope this definitive empirical evidence fully resolves your remaining concern.

---

### Official Review · Reviewer_eQYy · 2026-03-13

**Soundness:** 3
**Presentation:** 3
**Significance:** 3
**Originality:** 3
**Overall Recommendation:** 4
**Confidence:** 3

**Summary:**

The paper proposes a method for continuous-time learning of unsupervised prototypes in time-series, with the goal of detecting network faillures, using Neural ODE.
Six scenarios are tested and results are competitive with traditional approaches.

**Compliance With Llm Reviewing Policy:**

Affirmed.

**Final Justification:**

The rebuttal addressed my original concerns about complexity, OOD, and limitations more substantively than I first expected, but broader concerns remain about the strength of the "adaptive optimal prototype discovery" claim and the relatively narrow empirical scope, as raised in the discussion. So, although my original questions are answered and the tools employed are interesting, I am keeping my score because of the aforementioned points raised in the discussion with other reviewers.

**Key Questions For Authors:**

see Strengths And Weaknesses.

**Limitations:**

There is no explicit limitations section, and this is an important aspect. What types of problems or data are completely outside the scope of the method? Is there a length of horizon where the method breaks?

**Strengths And Weaknesses:**

## Strengths

Interesting and novel combination of previous methods applied to a specific and important problem.

## Weaknesses

Computational complexity was not assessed, and OOD behaviour, e.g., exploration of how the method woulf fare in some other dataset or problem was not explored

---

> ### Author Rebuttal · Authors · 2026-03-30
>
> We sincerely thank the reviewer for recognizing the novelty and significance of our approach.
>
> **Q1: Assessment of computational complexity.**
>
> We will add a theoretical complexity comparison in the Appendix. To ensure fairness, all baselines maintain comparable parameter complexity $\mathcal{O}(|\theta|)$, proving gains stem from continuous-time modeling, not parameter inflation.
>
> - Training Memory: Discrete models store intermediate states for backpropagation, costing $\mathcal{O}(L \cdot d_h)$ ($L$: sequence length, $d_h$: hidden dimension). H-ODE employs the adjoint sensitivity method, decoupling memory from integration steps to bound it at $\mathcal{O}(d_h)$. The memory complexity is $\mathcal{O}(1)$ with respect to $L$, preventing out-of-memory issues for long-horizon signals.
>
> - Training Time: This memory efficiency trades off with training time. Discrete models execute $L$ mappings, costing $\mathcal{O}(L \cdot C_f)$, where $C_f$ is the computational cost of a single forward evaluation. H-ODE's solver adaptively adjusts step sizes to meet error tolerances, yielding $\mathcal{O}(N_{eval} \cdot C_f)$, where $N_{eval}$ is the total number of function evaluations. Since typically $N_{eval} > L$, offline training requires more time.
>
> - Inference: Online inference triggers only a single forward integration without the adjoint backward pass. The overhead is deterministic, satisfying real-time requirements for proactive handovers.
>
> **Q2: OOD behavior and generalization capability.**
>
> While focused on the communication domain, our real-world dataset inherently presents severe OOD challenges via unforeseeable physical occlusions (e.g., dynamic crowds, structural changes). Our robustness against unseen disturbances stems from the continuous inductive bias. Discrete models fit statistical mappings, easily overfitting to local OOD noise. Conversely, our integral formulation forces latent states to follow a smooth derivative evolution. This acts as a structural stabilizer, effectively decoupling transient stochastic OOD noise from true physical degradation prototypes. A detailed discussion will be added.
>
> **Q3: Explicit limitations regarding out-of-scope data and horizon length.**
>
> We will add a "Limitations" section before the conclusion:
>
> - Out-of-scope data: Our framework mathematically assumes time-series evolution is driven by underlying continuous physical dynamics. Sequences lacking continuous inertia or dominated by pure discrete logic (e.g., text symbols, high-frequency financial ticks) fall outside our design scope.
>
> - Horizon limits: The ODE generative decoder encounters a boundary during extremely long unobserved windows. Without external observations $x_i$ to trigger GRU-based state rectification, pure numerical integration accumulates errors, eventually causing trajectory drift. Practical deployment requires intermittent updates within a confidence window.
>
> **Additional Validation Experiments.**
>
> To further solidify our claims, we conducted three comprehensive supplementary validation experiments during this rebuttal phase (available at https://anonymous.4open.science/r/Figure-D10C). First, comparisons against ModernTCN (Luo & Wang, 2024) (Figs. 1-4, Table 1) prove our continuous ODE prior fundamentally outperforms highly engineered discrete architectures in downstream FAR and FRR. Second, controlled reconstructions of step/smooth signals (Fig. 5) visually confirm the ODE's natural denoising capability against stochastic noise. Finally, replacing our adaptive module with standard K-means (Fig. 6) demonstrates that forcing rigid predefined $K$ values severely degrades system performance, firmly validating our data-driven $\tau$ mechanism.
>
>
> **Ref:** Luo & Wang. ModernTCN: A modern pure convolution structure for general time series analysis. ICLR 2024.

---

> > ### Author Rebuttal · Reviewer_eQYy · 2026-04-03
> >
> > I thank the Authors for the detailed answer.
> >
> > **Computational complexity**
> > As someone who worked with NeuralODE, I know that training times can be much larger than for non-ODE NN, but the benefits can also be large, depending on the problem. This comparison of the complexity of the proposed NODE and a regular NN should be clearly stated to help researchers who have never worked with these models understand the trade-offs.
> >
> > Thus, adding a table comparing NODEs and regular NNs in terms of complexity is in order.
> >
> > OOD behavior should be explained and tested. Occlusions and disturbances in the same environment do not, for this Reviewer, qualify as an OOD situation. Imagine a building scenario where you remove one floor, because the office is reduced in size. Or the office relocates and the system must be carried over. There will be a degradation, for sure. But can the degradation be quantified when comparing to a regular NN?

---

> > > ### Author Response · Authors · 2026-04-04
> > >
> > > **Assessment of computational complexity**
> > >
> > > We thank the reviewer for their comment regarding the computational tradeoffs of Neural ODEs. We have added a comprehensive theoretical complexity comparison to the revised Appendix (Table 1: https://anonymous.4open.science/r/Figure-1-2-C71B).
> > >
> > > Let $L$ denote sequence length, $d_h$ hidden dimension, $d$ input dimension, $k$ kernel size, and $N$/$\tilde{N}$ denote ODE solver evaluations during training/inference.
> > >
> > > * **Parameter Complexity:** All evaluated architectures (LSTM/GRU, ModernTCN, ResNet, ODE) are bounded at $\mathcal{O}(d_h^2)$, ensuring performance gains stem from continuous modeling rather than parameter inflation.
> > > * **Memory Complexity:** Discrete models require $\mathcal{O}(L \cdot d_h)$ to cache intermediate states. Using the adjoint sensitivity method, the ODE reconstructs trajectories backward, achieving $\mathcal{O}(1)$ memory with respect to $L$ and $N$, preventing out-of-memory issues for long signals.
> > > * **Training Time:** This memory efficiency introduces a tradeoff. Discrete models execute $L$ mappings ($\mathcal{O}(L \cdot d_h^2)$). The ODE performs $N$ sequential evaluations. Since $N > L$ for high-resolution dynamics, offline training complexity increases to $\mathcal{O}(N \cdot d_h^2)$.
> > > * **Inference Time:** Online execution takes $\tilde{N}$ steps ($\mathcal{O}(\tilde{N} \cdot d_h^2)$) for ODE. While theoretically higher, this forward-only overhead remains deterministic and lightweight, satisfying real-time proactive handover requirements.
> > >
> > > **Assessment of OOD behavior and quantified degradation**
> > >
> > > We thank the reviewer for noting that true OOD testing requires spatial relocation. To address this, we conducted a "Cross-Environment Evaluation."
> > >
> > > **1. Rationale of the OOD Setup**
> > > Our design focuses on "Intent Invariance." While the movement intent (e.g., transitioning to a target boundary) remains consistent, the spatial trajectories and radio environments change completely. This creates a domain shift as the system relocates to an unseen building with unique multipath signatures. We use a rigorous zero-shot protocol: both model weights and recognition templates are derived exclusively from the source scenario. This represents an extreme worst-case evaluation, as absolutely no data from the new environment is involved in incremental learning or model updating. In practice, our system rarely faces such strict limitations due to its rapid adaptation capabilities.
> > >
> > > **2. Quantified Degradation Analysis**
> > > Table 2 (https://anonymous.4open.science/r/Figure-1-2-C71B) highlights the performance gap upon relocation. (*Intra-Env* is the baseline tested in the training environment; *OOD-Trans* is a zero-shot test transferring the source-trained model and templates to an unseen environment.) Specifically, we performed cross-evaluations between Scenario 2 and Scenario 3.
> > > * **Failure of Regular NNs:** Discrete models rely on environment-specific fingerprints. Upon relocation, their representations fail to map, causing a performance collapse (FRR > 0.90).
> > > * **Generalization of ODE:** The continuous ODE maintains stable performance by capturing invariant physical laws, such as continuous velocity constraints.
> > > * **Explanation of Shift:** The FRR increase during OOD-Trans reflects the "Spatial Relocation Degradation" noted by the reviewer, caused by drastic multipath changes shifting the latent data distribution.
> > >
> > > **3. Practical Adaptation**
> > > Although zero-shot testing is a stress test, our system supports rapid adaptation:
> > > * **Efficiency:** Embedded physical constraints allow the ODE to converge in new scenarios with fewer samples than regular NNs.
> > > * **Fast Update:** Templates can be updated using minimal new-environment data, ensuring efficient carry-over.

---

### Decision · Program_Chairs · 2026-04-30

**Decision:**

Accept (regular)

**Comment:**

The most significant contribution is the continuous-time neural ODE formulation for prototype learning, which reviewers found to be a coherent and well-motivated approach that demonstrably separates physical signal degradation from stochastic noise in irregularly sampled wireless time series and avoids predefining the cluster count K.

Reviewers converged on the strengths of the adjoint-based memory-efficient training, the adaptive hierarchical prototype discovery, and the FAR/FRR gains over discrete architectures. There are some minor concerns about the limited novelty arising from combining existing components without a fundamentally new algorithmic primitive, about the single-domain evaluation that constrains generality claims. Authors are expected in the final version to incorporate the supplementary experiments and complexity tables provided during the rebuttal, add the promised limitations section addressing τ-base sensitivity and integration drift over long unobserved windows, describe the six experimental scenarios and the FAR/FRR classification protocol explicitly in the main text, and include FAR/FRR results for the sparsity experiment to complete the empirical picture.